# Automatic Data Augmentation for Generalization in Reinforcement Learning

## Abstract

Deep reinforcement learning (RL) agents often fail to generalize beyond their training environments. To alleviate this problem, recent work has proposed the use of data augmentation. However, different tasks tend to benefit from different types of augmentations and selecting the right one typically requires expert knowledge. In this paper, we introduce three approaches for automatically finding an effective augmentation for any RL task. These are combined with two novel regularization terms for the policy and value function, required to make the use of data augmentation theoretically sound for actor-critic algorithms. We evaluate our method on the Procgen benchmark which consists of 16 procedurally generated environments and show that it improves test performance by $40\%$ relative to standard RL algorithms. Our approach also outperforms methods specifically designed to improve generalization in RL, thus setting a new state-of-the-art on Procgen. In addition, our agent learns policies and representations which are more robust to changes in the environment that are irrelevant for solving the task, such as the background.

## 1 Introduction

Generalization to new environments remains a major challenge in deep reinforcement learning (RL). Current methods fail to generalize to unseen environments even when trained on similar settings (Farebrother et al., 2018; Packer et al., 2018; Zhang et al., 2018a; Cobbe et al., 2018; Gamrian & Goldberg, 2019; Cobbe et al., 2019; Song et al., 2020). This indicates that standard RL agents memorize specific trajectories rather than learning transferable skills. Several strategies have been proposed to alleviate this problem, such as the use of regularization (Farebrother et al., 2018; Zhang et al., 2018a; Cobbe et al., 2018; Igl et al., 2019), data augmentation (Cobbe et al., 2018; Lee et al., 2020; Ye et al., 2020; Kostrikov et al., 2020; Laskin et al., 2020), or representation learning (Zhang et al., 2020a;c). In this work, we focus on the use of data augmentation in RL. We identify key differences between supervised learning and reinforcement learning which need to be taken into account when using data augmentation in RL.

More specifically, we show that a naive application of data augmentation can lead to both theoretical and practical problems with standard RL algorithms, such as unprincipled objective estimates and poor performance. As a solution, we propose **Data-regularized Actor-Critic** or **DrAC**, a new algorithm that enables the use of data augmentation with actor-critic algorithms in a theoretically sound way. Specifically, we introduce two regularization terms which constrain the agent's policy and value function to be invariant to various state transformations. Empirically, this approach allows the agent to learn useful behaviors (outperforming strong RL baselines) in settings in which a naive use of data augmentation completely fails or converges to a sub-optimal policy. While we use Proximal Policy Optimization (PPO, Schulman et al. (2017)) to describe and validate our approach, the method can be easily integrated with any actor-critic algorithm with a discrete stochastic policy such as A3C (Mnih et al., 2013), SAC (Haarnoja et al., 2018), or IMPALA (Espeholt et al., 2018).

The current use of data augmentation in RL either relies on expert knowledge to pick an appropriate augmentation (Cobbe et al., 2018; Lee et al., 2020; Kostrikov et al., 2020) or separately evaluates a large number of transformations to find the best one (Ye et al., 2020; Laskin et al., 2020). In this paper, we propose three methods for automatically finding a useful augmentation for a given RL task. The first two learn to select the best augmentation from a fixed set, using either a variant of the upper confidence bound algorithm (UCB, Auer (2002)) or meta-learning ($RL^2$, Wang et al. (2016)).

Figure 1: Overview of UCB-DrAC. A UCB bandit selects an image transformation (*e.g.* random-conv) and applies it to the observations. The augmented and original observations are passed to a regularized actor-critic agent (*i.e.* DrAC) which uses them to learn a policy and value function which are invariant to this transformation.

We refer to these methods as **UCB-DrAC** and **RL2-DrAC**, respectively. The third method, **Meta-DrAC**, directly meta-learns the weights of a convolutional network, without access to predefined transformations (MAML, Finn et al. (2017)). Figure 1 gives an overview of UCB-DrAC.

We evaluate these approaches on the *Procgen* generalization benchmark (Cobbe et al., 2019) which consists of 16 procedurally generated environments with visual observations. Our results show that UCB-DrAC is the most effective among these at finding a good augmentation, and is comparable or better than using DrAC with the best augmentation from a given set. UCB-DrAC also outperforms baselines specifically designed to improve generalization in RL (Igl et al., 2019; Lee et al., 2020; Laskin et al., 2020) on both train and test. In addition, we show that our agent learns policies and representations that are more invariant to changes in the environment which do not alter the reward or transition function (*i.e.* they are inconsequential for control), such as the background theme.

To summarize, our work makes the following contributions: (i) we introduce a principled way of using data augmentation with actor-critic algorithms, (ii) we propose a practical approach for automatically selecting an effective augmentation in RL settings, (iii) we show that the use of data augmentation leads to policies and representations that better capture task invariances, and (iv) we demonstrate state-of-the-art results on the Procgen benchmark.

## 2   BACKGROUND

We consider a distribution $q(m)$ of Markov decision processes (MDPs, Bellman (1957)) $m \in \mathcal{M}$, with $m$ defined by the tuple $(\mathcal{S}_m, \mathcal{A}, T_m, R_m, p_m, \gamma)$, where $\mathcal{S}_m$ is the state space, $\mathcal{A}$ is the action space, $T_m(s'|s, a)$ is the transition function, $R_m(s, a)$ is the reward function, and $p_m(s_0)$ is the initial state distribution. During training, we restrict access to a fixed set of MDPs, $M_{train} = \{m_1, \ldots, m_n\}$, where $m_i \sim q, \forall i = \overline{1, n}$. The goal is to find a policy $\pi_\theta$ which maximizes the expected discounted reward over the entire distribution of MDPs, $J(\pi_\theta) = \mathbb{E}_{q,\pi,T_m,p_m} \left[ \sum_{t=0}^{T} \gamma^t R_m(s_t, a_t) \right]$.

In practice, we use the Procgen benchmark which contains 16 procedurally generated games. Each game corresponds to a distribution of MDPs $q(m)$, and each level of a game corresponds to an MDP sampled from that game's distribution $m \sim q$. The MDP $m$ is determined by the seed (*i.e.* integer) used to generate the corresponding level. Following the setup from Cobbe et al. (2019), agents are trained on a fixed set of $n = 200$ levels (generated using seeds from 1 to 200) and tested on the full distribution of levels (generated by sampling seeds uniformly at random from all computer integers).

**Proximal Policy Optimization** (PPO, Schulman et al. (2017)) is an actor-critic algorithm that learns a policy $\pi_\theta$ and a value function $V_\theta$ with the goal of finding an optimal policy for a given MDP. PPO alternates between sampling data through interaction with the environment and maximizing a clipped surrogate objective function $J_{\mathrm{PPO}}$ using stochastic gradient ascent. See Appendix A for a full description of PPO. One component of the PPO objective is the policy gradient term $J_{\mathrm{PG}}$, which is estimated using importance sampling:

$$J_{\mathrm{PG}}(\theta) = \sum_{a \in \mathcal{A}} \pi_\theta(a|s) \hat{A}_{\theta_{\mathrm{old}}}(s, a) = \mathbb{E}_{a \sim \pi_{\theta_{\mathrm{old}}}} \left[ \frac{\pi_\theta(a|s)}{\pi_{\theta_{\mathrm{old}}}(a|s)} \hat{A}_{\theta_{\mathrm{old}}}(s, a) \right], \tag{1}$$

where $\hat{A}(\cdot)$ is an estimate of the advantage function, $\pi_{\theta_{\mathrm{old}}}$ is the behavior policy used to collect trajectories (*i.e.* that generates the training distribution of states and actions), and $\pi_\theta$ is the policy we want to optimize (*i.e.* that generates the true distribution of states and actions).

## 3    Automatic Data Augmentation for RL

### 3.1    Data Augmentation in RL

Image augmentation has been successfully applied in computer vision for improving generalization on object classification tasks (Simard et al., 2003; Cireşan et al., 2011; Ciregan et al., 2012; Krizhevsky et al., 2012). As noted by Kostrikov et al. (2020), those tasks are invariant to certain image transformations such as rotations or flips, which is not always the case in RL. For example, if your observation is flipped, the corresponding reward will be reversed for the left and right actions and will not provide an accurate signal to the agent. While data augmentation has been previously used in RL settings without other algorithmic changes (Cobbe et al., 2018; Ye et al., 2020; Laskin et al., 2020), we argue that this approach is not theoretically sound.

If transformations are naively applied to observations in PPO's buffer, as done in Laskin et al. (2020), the PPO objective changes and equation (1) is replaced by

$$J_{\mathrm{PG}}(\theta) = \sum_{a \in \mathcal{A}} \pi_\theta(a|s) \hat{A}_{\theta_{\mathrm{old}}}(s,a) = \mathbb{E}_{a \sim \pi_{\theta_{\mathrm{old}}}} \left[ \frac{\pi_\theta(a|f(s))}{\pi_{\theta_{\mathrm{old}}}(a|s)} \hat{A}_{\theta_{\mathrm{old}}}(s,a) \right], \tag{2}$$

where $f : \mathcal{S} \times \mathcal{H} \to \mathcal{S}$ is the image transformation. However, the right hand side of the above equation is not a sound estimate of the left hand side because $\pi_\theta(a|f(s)) \neq \pi_\theta(a|s)$, since nothing constrains $\pi_\theta(a|f(s))$ to be close to $\pi_\theta(a|s)$. Moreover, one can define certain transformations $f(\cdot)$ that result in an arbitrarily large ratio $\pi_\theta(a|f(s))/\pi_\theta(a|s)$.

Figure 2 shows examples where a naive use of data augmentation prevents PPO from learning a good policy in practice, suggesting that this is not just a theoretical concern. In the following section, we propose an algorithmic change that enables the use of data augmentation with actor-critic algorithms in a principled way.

### 3.2    Policy and Value Function Regularization

Inspired by the recent work of Kostrikov et al. (2020), we propose two novel regularization terms for the policy and value functions that enable the proper use of data augmentation for actor-critic algorithms. Our algorithmic contribution differs from that of Kostrikov et al. (2020) in that it constrains both the actor and the critic, as opposed to only regularizing the Q-function.

Following Kostrikov et al. (2020), we define an optimality-invariant state transformation $f : \mathcal{S} \times \mathcal{H} \to \mathcal{S}$ as a mapping that preserves both the agent's policy $\pi$ and its value function $V$ such that $V(s) = V(f(s,\nu))$ and $\pi(a|s) = \pi(a|f(s,\nu))$, $\forall s \in \mathcal{S}$, $\nu \in \mathcal{H}$, where $\nu$ are the parameters of $f(\cdot)$, drawn from the set of all possible parameters $\mathcal{H}$.

To ensure that the policy and value functions are invariant to such transformation of the input state, we propose an additional loss term for regularizing the policy,

$$G_\pi = KL\left[\pi_\theta(a|s) \mid \pi_\theta(a|f(s,\nu))\right], \tag{3}$$

as well as an extra loss term for regularizing the value function,

$$G_V = \left(V_\phi(s) - V_\phi\left(f(s,\nu)\right)\right)^2. \tag{4}$$

Thus, our **data-regularized actor-critic** method, or **DrAC**, maximizes the following objective:

$$J_{\mathrm{DrAC}} = J_{\mathrm{PPO}} - \alpha_r(G_\pi + G_V), \tag{5}$$

where $\alpha_r$ is the weight of the regularization term (see Algorithm 1).

The use of $G_\pi$ and $G_V$ ensures that the agent's policy and value function are invariant to the transformations induced by various augmentations. Particular transformations can be used to impose certain inductive biases relevant for the task (*e.g.* invariance with respect to colors or translations). In addition, $G_\pi$ and $G_V$ can be added to the objective of any actor-critic algorithm with a discrete stochastic policy (*e.g.* A3C, TRPO, ACER, SAC, or IMPALA) without any other changes.

Note that when using DrAC, as opposed to the method proposed by Laskin et al. (2020), we still use the correct importance sampling estimate of the left hand side objective in equation (1) (instead of a

wrong estimate as in equation (2)). This is because the transformed observations $f(s)$ are only used to compute the regularization losses $G_\pi$ and $G_V$, and thus are not used for the main PPO objective. Without these extra terms, the only way to use data augmentation is as explained in Section 3.1, which leads to inaccurate estimates of the PPO objective. Hence, DrAC benefits from the regularizing effect of using data augmentation, while mitigating adverse consequences on the RL objective. See Appendix B for a more detailed explanation of why a naive application of data augmentation with certain policy gradient algorithms is theoretically unsound.

---

**Algorithm 1 DrAC**: **D**ata-**r**egularized **A**ctor-**C**ritic applied to PPO
Black: unmodified actor-critic algorithm.
Cyan: image transformation.
Red: policy regularization.
Blue: value function regularization.

---

1: **Hyperparameters:** image transformation $f$, regularization loss coefficient $\alpha_r$, minibatch size M, replay buffer size T, number of updates K.
2: **for** $k = 1, \ldots, K$ **do**
3:      Collect a new set of transitions $\mathcal{D} = \{(s_i, a_i, r_i, s_{i+1})\}_{i=1}^T$ using $\pi_\theta$.
4:      **for** $j = 1, \ldots, \lfloor \frac{T}{M} \rfloor$ **do**
5:          $\{(s_i, a_i, r_i, s_{i+1})\}_{i=1}^M \sim \mathcal{D}$                 $\triangleright$ Sample a minibatch of transitions
6:          **for** $i = 1, \ldots, M$ **do**
7:              $\nu_i \sim \mathcal{H}$                          $\triangleright$ Sample the augmentation parameters
8:              $\hat{\pi}_i \leftarrow \pi_\phi(\cdot | s_i)$                    $\triangleright$ Compute the policy targets
9:              $\hat{V}_i \leftarrow V_\phi(s_i)$                  $\triangleright$ Compute the value function targets
10:          **end for**
11:          $G_\pi(\theta) = \frac{1}{M} \sum_{i=1}^M KL[\hat{\pi}_i \mid \pi_\theta(\cdot | f(s_i, \nu_i))]$          $\triangleright$ Regularize the policy
12:          $G_V(\phi) = \frac{1}{M} \sum_{i=1}^M \left( \hat{V}_i - V_\phi(f(s_i, \nu_i)) \right)^2$       $\triangleright$ Regularize the value function
13:          $J_{\text{DrAC}}(\theta, \phi) = J_{\text{PPO}}(\theta, \phi) - \alpha_r(G_\pi(\theta) + G_V(\phi))$    $\triangleright$ Compute the total loss function
14:          $\theta \leftarrow \arg\max_\theta J_{\text{DrAC}}$                     $\triangleright$ Update the policy
15:          $\phi \leftarrow \arg\max_\phi J_{\text{DrAC}}$                    $\triangleright$ Update the value function
16:      **end for**
17: **end for**

---

## 3.3 AUTOMATIC DATA AUGMENTATION

Since different tasks benefit from different types of transformations, we would like to design a method that can automatically find an effective transformation for any given task. Such a technique would significantly reduce the computational requirements for applying data augmentation in RL. In this section, we describe three approaches for doing this. In all of them, the augmentation learner is trained at the same time as the agent learns to solve the task using DrAC. Hence, the distribution of rewards varies significantly as the agent improves, making the problem highly nonstationary.

**Upper Confidence Bound.** The problem of selecting a data augmentation from a given set can be formulated as a multi-armed bandit problem, where the action space is the set of available transformations $\mathcal{F} = \{f^1, \ldots, f^n\}$. A popular algorithm for such settings is the upper confidence bound or UCB (Auer, 2002), which selects actions according to the following policy:

$$f_t = \text{argmax}_{f \in \mathcal{F}} \left[ Q_t(f) + c \sqrt{\frac{\log(t)}{N_t(f)}} \right], \tag{6}$$

where $f_t$ is the transformation selected at time step $t$, $N_t(f)$ is the number of times transformation $f$ has been selected before time step $t$ and $c$ is UCB's exploration coefficient. Before the t-th DrAC update, we use equation (6) to select an augmentation $f$. Then, we use equation (5) to update the agent's policy and value function. We also update the counter: $N_t(f) = N_{t-1}(f) + 1$. Next, we collect rollouts with the new policy and update the Q-function: $Q_t(f) = \frac{1}{K} \sum_{i=t-K}^t \mathcal{R}(f_i = f)$, which is computed as a sliding window average of the past $K$ mean returns obtained by the agent after being updated using augmentation $f$. We refer to this algorithm as **UCB-DrAC** (Algorithm 4).

Note that UCB-DrAC's estimation of $Q(f)$ differs from that of a typical UCB algorithm which uses rewards from the entire history. However, the choice of estimating $Q(f)$ using only more recent rewards is crucial due to the nonstationarity of the problem.

**Meta-Learning the Selection of an Augmentation.** Alternatively, the problem of selecting a data augmentation from a given set can be formulated as a meta-learning problem. Here, we consider a meta-learner like the one proposed by Wang et al. (2016). Before each DrAC update, the meta-learner selects an augmentation, which is then used to update the agent using equation (5). We then collect rollouts using the new policy and update the meta-learner using the mean return of these trajectories. We refer to this approach as **RL2-DrAC** (Algorithm 4).

**Meta-Learning the Weights of an Augmentation.** Another approach for automatically finding an appropriate augmentation is to directly learn the weights of a certain transformation rather than selecting an augmentation from a given set. In this work, we focus on meta-learning the weights of a convolutional network which can be applied to the observations to obtain a perturbed image. We meta-learn the weights of this network using an approach similar to the one proposed by Finn et al. (2017). For each agent update, we also perform a meta-update of the transformation function by splitting DrAC's buffer into meta-train and meta-test sets. We refer to this approach as **Meta-DrAC** (Algorithm 4). More details about the implementation of these methods can be found in Appendix D.

## 4 Experiments

In this section, we evaluate our methods on the Procgen benchmark (Cobbe et al., 2019) which consists of 16 procedurally generated games (see Figure 6 in Appendix G). Procgen has a number of attributes that make it a good testbed for generalization in RL: (i) it has a diverse set of games in a similar spirit with the ALE benchmark (Bellemare et al., 2013), (ii) each of these games has procedurally generated levels which present agents with meaningful generalization challenges, (iii) agents have to learn motor control directly from images, and (iv) it has a clear protocol for testing generalization.

All environments use a discrete 15 dimensional action space and produce $64 \times 64 \times 3$ RGB observations. We use Procgen's *easy* setup, so for each game, agents are trained on 200 levels and tested on the full distribution of levels. We use PPO as a base for all our methods. More details about our experimental setup and hyperparameters can be found in Appendix E.

**Data Augmentation.** In our experiments, we use a set of eight transformations: *crop, grayscale, cutout, cutout-color, flip, rotate, random convolution* and *color-jitter* (Krizhevsky et al., 2012; DeVries & Taylor, 2017). We use **RAD**'s (Laskin et al., 2020) implementation of these transformations, except for *crop*, in which we pad the image with 12 (boundary) pixels on each side and select random crops of $64 \times 64$. We found this implementation of *crop* to be significantly better on Procgen, and thus it can be considered an empirical upper bound of RAD in this case. For simplicity, we will refer to our implementation as RAD. **DrAC** uses the same set of transformations as RAD, but is trained with additional regularization losses for the actor and the critic, as described in Section 3.2.

**Automatic Selection of Data Augmentation.** We compare three different approaches for automatically finding an effective transformation: **UCB-DrAC** which uses UCB (Auer, 2002) to select an augmentation from a given set, **RL2-DrAC** which uses $\text{RL}^2$ (Wang et al., 2016) to do the same, and **Meta-DrAC** which uses MAML (Finn et al., 2017) to meta-learn the weights of a convolutional network. Meta-DrAC is implemented using the *higher* library (Grefenstette et al., 2019). Note that we do not expect these approaches to be better than DrAC with the best augmentation. In fact, DrAC with the best augmentation can be considered to be an upper bound for these automatic approaches since it uses the best augmentation during the entire training process.

**Ablations.** **Rand-DrAC** uses a uniform distribution to select an augmentation each time. **Crop-DrAC** uses crop for all games (which is the most effective augmentation on half of the Procgen games). **UCB-RAD** combines UCB with RAD (*i.e.* it does not use the regularization terms).

**Baselines.** We also compare with **Rand-FM** (Lee et al., 2020) and **IBAC-SNI** (Igl et al., 2019), two methods specifically designed for improving generalization in RL and previously tested on CoinRun, one of the Procgen games. Rand-FM uses a random convolutional networks to regularize the learned representations, while IBAC-SNI uses an information bottleneck with selective noise injection.

Table 1: Train and test performance for the Procgen benchmark (aggregated over all 16 tasks, 10 seeds). (a) compares PPO with two baselines specifically designed to improve generalization in RL and shows that they do not significantly help. (b) compares using the best augmentation from our set with and without regularization, corresponding to DrAC and RAD respectively, and shows that regularization improves performance on both train and test. (c) compares different approaches for automatically finding an augmentation for each task, namely using UCB or $RL^2$ for selecting the best transformation from a given set, or meta-learning the weights of a convolutional network (Meta-DrAC). (d) shows additional ablations: Rand-DrAC selects an augmentation using a uniform distribution, Crop-DrAC uses image crops for all tasks, and UCB-RAD is an ablation that does not use the regularization losses. UCB-DrAC performs best on both train and test, and achieves a return comparable with or better than DrAC (which uses the best augmentation).

| | | PPO-Normalized Return (%) | | | | | |
| | | Train | | | Test | | |
| | **Method** | **Median** | **Mean** | **Std** | **Median** | **Mean** | **Std** |
|---|---|---|---|---|---|---|---|
| (a) | PPO | 100.0 | 100.0 | 7.2 | 100.0 | 100.0 | 8.5 |
| | Rand-FM | 93.4 | 87.6 | 8.9 | 91.6 | 78.0 | 9.0 |
| | IBAC-SNI | 91.9 | 103.4 | 8.5 | 86.2 | 102.9 | 8.6 |
| (b) | DrAC (Best) | **114.0** | **119.6** | 9.4 | 118.5 | 138.1 | 10.5 |
| | RAD (Best) | 103.7 | 109.1 | 9.6 | 114.2 | 131.3 | 9.4 |
| (c) | UCB-DrAC (Ours) | 102.3 | 118.9 | 8.8 | **118.5** | **139.7** | 8.4 |
| | RL2-DrAC | 96.3 | 95.0 | 8.8 | 99.1 | 105.3 | 7.1 |
| | Meta-DrAC | 101.3 | 100.1 | 8.5 | 101.7 | 101.2 | 7.3 |
| (d) | Rand-DrAC | 100.4 | 99.5 | 8.4 | 102.4 | 103.4 | 7.0 |
| | Crop-DrAC | 97.4 | 112.8 | 9.8 | 114.0 | 132.7 | 11.0 |
| | UCB-RAD | 100.4 | 104.8 | 8.4 | 103.0 | 125.9 | 9.5 |

**Evaluation Metrics.** At the end of training, for each method and each game, we compute the average score over 100 episodes and 10 different seeds. The scores are then normalized using the corresponding PPO score on the same game. We aggregate the normalized scores over all 16 Procgen games and report the resulting mean, median, and standard deviation (Table 1). For a per-game breakdown, see Tables 6 and 7 in Appendix I.

## 4.1 GENERALIZATION ABILITY

Table 1 shows train and test performance on Procgen. UCB-DrAC significantly outperforms PPO, Rand-FM, and IBAC-SNI. Regularizing the policy and value function leads to improvements over merely using data augmentation, and thus the performance of DrAC is better than that of RAD (both using the best augmentation for each game). Our experiments show that the most effective way of automatically finding an augmentation is UCB-DrAC. As expected, meta-learning the weights of a CNN using Meta-DrAC performs reasonably well on the games in which the random convolution augmentation helps. But overall, Meta-DrAC and RL2-DrAC are worse than UCB-DrAC. In addition, UCB is generally more stable, easier to implement, and requires less fine-tuning compared to meta-learning algorithms. See Figures 7 and 8 in Appendix J for a comparison of these three approaches on each game. Moreover, automatically selecting the augmentation from a given set using UCB-DrAC performs similarly well or even better than a method that uses the best augmentation for each task throughout the entire training process. UCB-DrAC also achieves higher returns than an ablation that uses a uniform distribution to select an augmentation each time, Rand-DrAC. Nevertheless, UCB-DrAC is better than Crop-DrAC, which uses crop for all the games (which is the best augmentation for eight of the Procgen games as shown in Tables 4 and 5 from Appendix H).

## 4.2 REGULARIZATION EFFECT

In Section 3.1, we argued that additional regularization terms are needed in order to make the use of data augmentation in RL theoretically sound. However, one might wonder if this problem actually appears in practice. Thus, we empirically investigate the effect of regularizing the policy and value

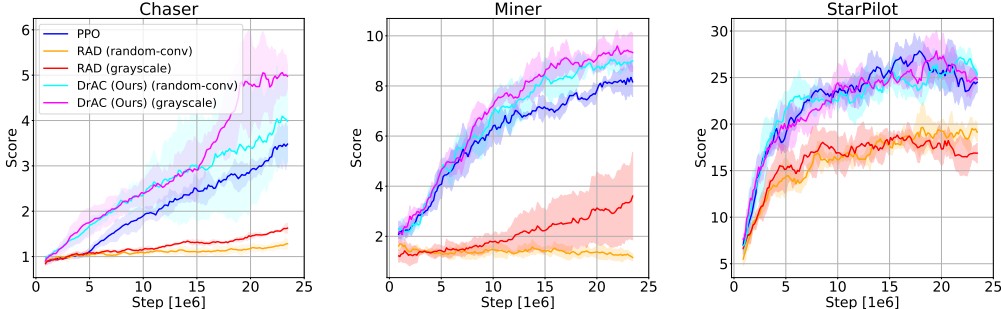

Figure 2: Comparison between RAD and DrAC with the same augmentations, grayscale and random convolution, on the test environments of Chaser (left), Miner (center), and StarPilot (right). While DrAC's performance is comparable or better than PPO's, not using the regularization terms, *i.e.* using RAD, significantly hurts performance relative to PPO. This is because, in contrast to DrAC, RAD does not use a principled (importance sampling) estimate of PPO's objective.

function. For this purpose, we compare the performance of RAD and DrAC with grayscale and random convolution augmentations on Chaser, Miner, and StarPilot.

Figure 2 shows that not regularizing the policy and value function with respect to the transformations used can lead to drastically worse performance than vanilla RL methods, further emphasizing the importance of these loss terms. In contrast, using the regularization terms as part of the RL objective (as DrAC does) results in an agent that is comparable or, in some cases, significantly better than PPO.

### 4.3 AUTOMATIC AUGMENTATION

Our experiments indicate there is not a single augmentation that works best across all Procgen games (see Tables 4 and 5 in Appendix H). Moreover, our intuitions regarding the best transformation for each game might be misleading. For example, at a first sight, Ninja appears to be somewhat similar to Jumper, but the augmentation that performs best on Ninja is color-jitter, while for Jumper is random-conv (see Tables 4 and 5). In contrast, Miner seems like a different type of game than Climber or Ninja, but they all have the same best performing augmentation, namely color-jitter. These observations further underline the need for a method that can automatically find the right augmentation for each task.

Table 1 along with Figures 7 and 8 in the Appendix compare different approaches for automatically finding an augmentation, showing that UCB-DrAC performs best and reaches the asymptotic performance obtained when the most effective transformation for each game is used throughout the entire training process. Figure 3 illustrates an example of UCB's policy during training on Ninja and Dodgeball, showing that it converges to always selecting the most effective augmentation, namely color-jitter for Ninja and crop for Dodgeball. Figure 5 in Appendix F illustrates how UCB's behavior varies with its exploration coefficient.

### 4.4 ROBUSTNESS ANALYSIS

To further investigate the generalizing ability of these agents, we analyze whether the learned policies and state representations are invariant to changes in the observations which are irrelevant for solving the task.

We first measure the Jensen-Shannon divergence (JSD) between the agent's policy for an observation from a training level and a modified version of that observation with a different background theme (*i.e.* color and pattern). Note that the JSD also represents a lower bound for the joint empirical risk across train and test (Ilse et al., 2020). The background theme is randomly selected from the set of backgrounds available for all other Procgen environments, except for the one of the original training level. Note that the modified observation has the same semantics as the original one (with respect to the reward function), so the agent should have the same policy in both cases. Note that many of the backgrounds are not uniform and can contain items such as trees or planets which can be easily

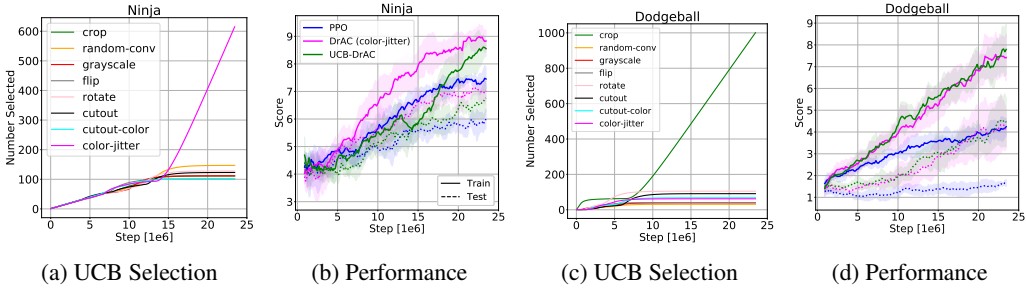

Figure 3: Cumulative number of times UCB selects each augmentation over the course of training for Ninja (a) and Dodgeball (c). Train and test performance for PPO, DrAC with the best augmentation for each game (color-jitter and crop, respectively), and UCB-DrAC for Ninja (b) and Dodgeball (d). UCB-DrAC finds the most effective augmentation from the given set and reaches the performance of DrAC. Our methods improve both train and test performance.

Table 2: JSD and Cycle-Consistency (%) (aggregated across all Procgen tasks) for PPO, RAD and UCB-DrAC, measured between observations that vary only in their background themes (*i.e.* colors and patterns that do not interact with the agent). UCB-DrAC learns more robust policies and representations that are more invariant to changes in the observation that are irrelevant for the task.

| | | | Cycle-Consistency (%) | | | |
| | **JSD** | | **2-way** | | **3-way** | |
| **Method** | **Mean** | **Median** | **Mean** | **Median** | **Mean** | **Median** |
| PPO | 0.25 | 0.23 | 20.50 | 18.70 | 12.70 | 5.60 |
| RAD | 0.19 | 0.18 | 24.40 | 22.20 | 15.90 | 8.50 |
| UCB-DrAC | **0.16** | **0.15** | **27.70** | **24.80** | **17.30** | **10.30** |

misled for objects the agent can interact with. As seen in Table 2, UCB-DrAC has a lower JSD than PPO, indicating that it learns a policy that is more robust to changes in the background.

To quantitatively evaluate the quality of the learned representation, we use the cycle-consistency metric proposed by Aytar et al. (2018) and also used by Lee et al. (2020). See Appendix C for more details about this metric. Table 2 reports the percentage of input observations in the seen environment that are cycle-consistent with trajectories in modified unseen environments, which have a different background but the same layout. UCB-DrAC has higher cycle-consistency than PPO, suggesting that it learns representations that better capture relevant task invariances.

## 4.5   DEEPMIND CONTROL SUITE EXPERIMENTS

In this section, we evaluate our approach on the DeepMind Control Suite from pixels (DMC, Tassa et al. (2018)). We use four tasks, namely Cartpole Balance, Finger Spin, Walker Walk, and Cheetah Run, in three settings with different types of backgrounds, namely the *default*, *simple* distractors, and *natural* videos from the Kinetics dataset (Kay et al., 2017), as introduced in Zhang et al. (2020b). See Figure 9 for a few examples. Note that in the simple and natural settings, the background is sampled from a list of videos at the beginning of each episode, which creates spurious correlations between the backgrounds and the rewards. In the simple and natural distractor settings, as shown in Figure 11, UCB-DrAC outperforms PPO and RAD with the best augmentation on all these environments in the most challenging setting with natural distractors. See Appendix K for results on DMC with default and simple distractor backgrounds, where our method also outperforms the baselines.

## 5   RELATED WORK

**Generalization in Deep RL.** A recent body of work has pointed out the problem of overfitting in deep RL (Rajeswaran et al., 2017; Machado et al., 2018; Packer et al., 2018; Zhang et al., 2018a;b; Cobbe et al., 2018; 2019; Yarats et al., 2019; Raileanu & Rocktäschel, 2020). A promising approach to prevent overfitting is to apply regularization techniques originally developed for supervised learning

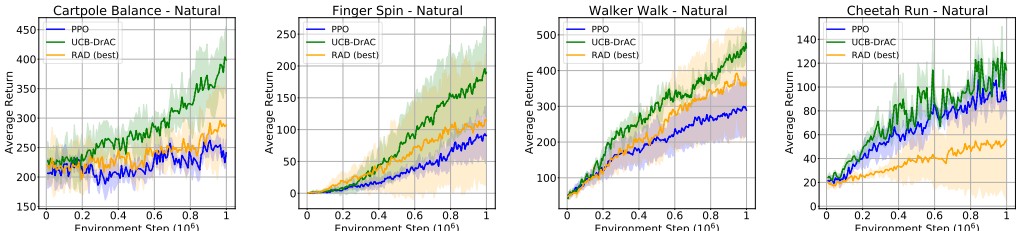

Figure 4: Average return on DMC tasks with natural video backgrounds with mean and standard deviation computed over 5 seeds. UCB-DrAC outperforms PPO and RAD with the best augmentations.

such as dropout (Srivastava et al., 2014) or batch normalization (Ioffe & Szegedy, 2015). Farebrother et al. (2018) and Cobbe et al. (2018) show that such regularization methods can improve the generalization ability of RL agents in Atari (Machado et al., 2018) and CoinRun (Cobbe et al., 2018), respectively. Similarly, Igl et al. (2019) use selective noise injection with a variational information bottleneck, while Lee et al. (2020) regularize the agent's representation with respect to random convolutional transformations. More recently, Sonar et al. (2020) learn invariant policies, Zhang et al. (2020a) and Zhang et al. (2020c) learn state abstractions using bisimulation, Roy & Konidaris (2020) align the features of two domains using Wasserstein distance, while Igl et al. (2020) reduce non-stationarity using policy distillation, and Mazoure et al. (2020) maximize the mutual information between the agent's internal representation of successive time steps. More similar to our work, Cobbe et al. (2018), Ye et al. (2020) and Laskin et al. (2020) add augmented observations to the training buffer of an RL agent. However, as we show here, naively applying data augmentation in RL can lead to both theoretical and practical issues. Our algorithmic contributions alleviate these problems while still benefitting from the regularization effect of data augmentation.

**Data Augmentation** has been extensively used in computer vision for both supervised (LeCun et al., 1989; Becker & Hinton, 1992; LeCun et al., 1998; Simard et al., 2003; Cireşan et al., 2011; Ciresan et al., 2011; Krizhevsky et al., 2012) and self-supervised (Dosovitskiy et al., 2016; Misra & van der Maaten, 2019) learning. More recent work uses data augmentation for contrastive learning, leading to state-of-the-art results on downstream tasks (Ye et al., 2019; Hénaff et al., 2019; He et al., 2019; Chen et al., 2020). Domain randomization can also be considered a type of data augmentation, which has proven useful for transferring RL policies from simulation to the real world (Tobin et al., 2017). However, domain randomization requires access to a physics simulator, which is not always available. Recently, a few papers propose the use of data augmentation in RL (Cobbe et al., 2018; Lee et al., 2020; Srinivas et al., 2020; Kostrikov et al., 2020; Laskin et al., 2020), but all of them use a fixed (set of) augmentation(s) rather than automatically finding the most effective one. The most similar work to ours is that of Kostrikov et al. (2020), who propose to regularize the Q-function in Soft Actor-Critic (SAC) (Haarnoja et al., 2018) using random shifts of the input image. Our work differs from theirs in that it automatically selects an augmentation from a given set, regularizes both the actor and the critic, and focuses on the problem of generalization rather than sample efficiency. While there is a body of work on the automatic use of data augmentation (Cubuk et al., 2019b;a; Fang et al., 2019; Shi et al., 2019; Li et al., 2020), these approaches were designed for supervised learning and, as we explain here, cannot be applied to RL without further algorithmic changes.

## 6 DISCUSSION

In this work, we propose UCB-DrAC, a method for automatically finding an effective data augmentation for RL tasks. Our approach enables the principled use of data augmentation with actor-critic algorithms by regularizing the policy and value functions with respect to state transformations. We show that UCB-DrAC avoids the theoretical and empirical pitfalls typical in naive applications of data augmentation in RL. Our approach improves training performance by $19\%$ and test performance by $40\%$ on the Procgen benchmark, and sets a new state-of-the-art on the Procgen benchmark. In addition, the learned policies and representations are more invariant to spurious correlations between observations and rewards. A promising avenue for future research is to use a more expressive function class for meta-learning the augmentations.

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

## A    PPO

**Proximal Policy Optimization** (PPO) Schulman et al. (2017) is an actor-critic RL algorithm that learns a policy $\pi_\theta$ and a value function $V_\theta$ with the goal of finding an optimal policy for a given MDP. PPO alternates between sampling data through interaction with the environment and optimizing an objective function using stochastic gradient ascent. At each iteration, PPO maximizes the following objective:

$$J_{\text{PPO}} = J_\pi - \alpha_1 J_V + \alpha_2 S_{\pi_\theta}, \tag{7}$$

where $\alpha_1$, $\alpha_2$ are weights for the different loss terms, $S_{\pi_\theta}$ is the entropy bonus for aiding exploration, $J_V$ is the value function loss defined as

$$J_V = \left(V_\theta(s) - V_t^{\text{target}}\right)^2.$$

The policy objective term $J_\pi$ is based on the policy gradient objective which can be estimated using importance sampling in off-policy settings (*i.e.* when the policy used for collecting data is different from the policy we want to optimize):

$$J_{PG}(\theta) = \sum_{a \in \mathcal{A}} \pi_\theta(a|s)\hat{A}_{\theta_{\text{old}}}(s,a) = \mathbb{E}_{a \sim \pi_{\theta_{\text{old}}}}\left[\frac{\pi_\theta(a|s)}{\pi_{\theta_{\text{old}}}(a|s)}\hat{A}_{\theta_{\text{old}}}(s,a)\right], \tag{8}$$

where $\hat{A}(\cdot)$ is an estimate of the advantage function, $\theta_{old}$ are the policy parameters before the update, $\pi_{\theta_{old}}$ is the behavior policy used to collect trajectories (*i.e.* that generates the training distribution of states and actions), and $\pi_\theta$ is the policy we want to optimize (*i.e.* that generates the true distribution of states and actions).

This objective can also be written as

$$J_{PG}(\theta) = \mathbb{E}_{a \sim \pi_{\theta_{\text{old}}}}\left[r(\theta)\hat{A}_{\theta_{\text{old}}}(s,a)\right], \tag{9}$$

where

$$r_\theta = \frac{\pi_\theta(a|s)}{\pi_{\theta_{\text{old}}}(a|s)}$$

is the importance weight for estimating the advantage function.

PPO is inspired by TRPO (Schulman et al., 2015), which constrains the update so that the policy does not change too much in one step. This significantly improves training stability and leads to better results than vanilla policy gradient algorithms. TRPO achieves this by minimizing the KL divergence between the old (*i.e.* before an update) and the new (*i.e.* after an update) policy. PPO implements the constraint in a simpler way by using a clipped surrogate objective instead of the more complicated TRPO objective. More specifically, PPO imposes the constraint by forcing $r(\theta)$ to stay within a small interval around 1, precisely $[1 - \epsilon, 1 + \epsilon]$, where $\epsilon$ is a hyperparameter. The policy objective term from equation (7) becomes

$$J_\pi = \mathbb{E}_\pi\left[\min\left(r_\theta\hat{A}, \ \text{clip}\left(r_\theta, 1 - \epsilon, 1 + \epsilon\right)\hat{A}\right)\right],$$

where $\hat{A} = \hat{A}_{\theta_{\text{old}}}(s,a)$ for brevity. The function $\text{clip}(r(\theta), 1 - \epsilon, 1 + \epsilon)$ clips the ratio to be no more than $1 + \epsilon$ and no less than $1 - \epsilon$. The objective function of PPO takes the minimum one between the original value and the clipped version so that agents are discouraged from increasing the policy update to extremes for better rewards.

Note that the use of the Adam optimizer (Kingma & Ba, 2015) allows loss components of different magnitudes so we can use $G_\pi$ and $G_V$ from equations (3) and (4) to be used as part of the DrAC objective in equation (5) with the same loss coefficient $\alpha_r$. This alleviates the burden of hyperparameter search and means that DrAC only introduces a single extra parameter $\alpha_r$.

## B    NAIVE APPLICATION OF DATA AUGMENTATION IN RL

In this section, we further clarify why a naive application of data augmentation with certain RL algorithms is theoretically unsound. This argument applies for all algorithms that use importance sampling for estimating the policy gradient loss, including TRPO, PPO, IMPALA, or ACER. The use of importance sampling is typically employed when the algorithm performs more than a single policy update using the same data in order to correct for the off-policy nature of the updates. For brevity, we will use PPO to explain this problem.

The correct estimate of the policy gradient objective used in PPO is the one in equation (1) (or equivalently, equation (8)) which does not use the augmented observations at all since we are estimating advantages for the actual observations, $A(s, a)$. The probability distribution used to sample advantages is $\pi_{old}(a|s)$ (rather than $\pi_{old}(a|f(s))$ since we can only interact with the environment via the true observations and not the augmented ones (because the reward and transition functions are not defined for augmented observations). Hence, the correct importance sampling estimate uses $\pi(a|s)/\pi_{old}(a|s)$. Using $\pi(a|f(s))/\pi_{old}(a|f(s))$ instead would be incorrect for the reasons mentioned above. What we argue is that, in the case of RAD, the only way to use the augmented observations $f(s)$ is in the policy gradient objective, whether by $\pi(a|f(s))/\pi_{old}(a|f(s))$ or $\pi(a|f(s))/\pi_{old}(a|s)$, depending on the exact implementation, but both of these are incorrect. In contrast, DrAC does not change the policy gradient objective at all which remains the one in equation (1) and instead uses the augmented observations in the additional regularization losses, as shown in equations (3), (4), and (5).

## C    CYCLE-CONSISTENCY

Here is a description of the cycle-consistency metric proposed by Aytar et al. (2018) and also used in Lee et al. (2020) for analyzing the learned representations of RL agents. Given two trajectories $V$ and $U$, $v_i \in V$ first locates its nearest neighbor in the other trajectory $u_j = \text{argmin}_{u \in U} \|h(v_i) - h(u)\|^2$, where $h(\cdot)$ denotes the output of the penultimate layer of trained agents. Then, the nearest neighbor of $u_j \in V$ is located, *i.e.*, $v_k = \text{argmin}_{v \in V} \|h(u_j) - h(u_j)\|_2$, and $v_i$ is defined as cycle-consistent if $|i - k| \leq 1$, *i.e.*, it can return to the original point. Note that this cycle-consistency implies that two trajectories are accurately aligned in the hidden space. Similar to Aytar et al. (2018), we also evaluate the three-way cycle-consistency by measuring whether vi remains cycle-consistent along both paths, $V \rightarrow U \rightarrow J \rightarrow V$ and $V \rightarrow J \rightarrow U \rightarrow V$, where J is the third trajectory.

## D    AUTOMATIC DATA AUGMENTATION ALGORITHMS

In this section, we provide more details about the automatic augmentation approaches, as well as pseudocodes for all three methods we propose.

RL2-DrAC uses an LSTM (Hochreiter & Schmidhuber, 1997) network to select an effective augmentation from a given set, which is used to update the agent's policy and value function according to the DrAC objective from Equation (5). We will refer to this network as a (recurrent) selection policy. The LSTM network takes as inputs the previously selected augmentation and the average return obtained after performing one update of the DrAC agent with this augmentation (using Algorithm 1 with K=1). The LSTM outputs an augmentation from the given set and is rewarded using the average return obtained by the agent after one update with the selected augmentation. The LSTM is trained using REINFORCE (Williams, 2004). See Algorithm 3 for a pseudocode of RL2-DrAC.

Meta-DrAC meta-learns the weights of a convolutional neural network (CNN) which is used to augment the observations in order to update the agent's policy and value function according to the DrAC objective from Equation (5). For each DrAC update of the agent, we split the trajectories in the replay buffer into meta-train and meta-test using a 9 to 1 ratio. The CNN's weights are updated using MAML (Finn et al., 2017) where the objective function is maximizing the average return obtained by DrAC (after an update using Algorithm 1 with $K = 1$ and the CNN as the transformation $f$).

---

**Algorithm 2 UCB-DrAC**

---

1: **Hyperparameters:** Set of image transformations $\mathcal{F} = \{f^1, \ldots, f^n\}$, exploration coefficient c, window for estimating the Q-functions W, number of updates K, initial policy parameters $\pi_\theta$, initial value function $V_\phi$.
2: $N(f) = 1, \forall f \in \mathcal{F}$       ▷ Initialize the number of times each augmentation was selected
3: $Q(f) = 0, \forall f \in \mathcal{F}$       ▷ Initialize the Q-functions for all augmentations
4: $R(f) = \text{FIFO}(W), \forall f \in \mathcal{F}$       ▷ Initialize the lists of returns for all augmentations
5: **for** $k = 1, \ldots, K$ **do**
6:      $f_k = \text{argmax}_{f \in \mathcal{F}} \left[ Q(f) + c \sqrt{\frac{\log(k)}{N(f)}} \right]$       ▷ Use UCB to select an augmentation
7:      Update the policy and value function according to Algorithm 1 with $f = f_k$ and $K = 1$:
8:      $\theta \leftarrow \arg\max_\theta J_{\text{DrAC}}$       ▷ Update the policy
9:      $\phi \leftarrow \arg\max_\phi J_{\text{DrAC}}$       ▷ Update the value function
10:      Compute the mean return obtained by the new policy $r_k$.
11:      Add $r_k$ to the $R(f_k)$ list using the first-in-first-out rule.
12:      $Q(f_k) \leftarrow \frac{1}{|R(f_k)|} \sum_{r \in R(f_k)} r$
13:      $N(f_k) \leftarrow N(f_k) + 1$
14: **end for**

---

---

**Algorithm 3 RL2-DrAC**

---

1: **Hyperparameters:** Set of image transformations $\mathcal{F} = \{f^1, \ldots, f^n\}$, number of updates K, initial policy $\pi_\theta$, initial value function $V_\phi$.
2: Initialize the selection poicy as an LSTM network $g$ with parameters $\psi$.
3: $f_0 \sim \mathcal{F}$       ▷ Randomly initialize the augmentation
4: $r_0 \leftarrow 0$       ▷ Initialize the average return
5: **for** $k = 1, \ldots, K$ **do**
6:      $f_k \sim g_\psi(f_{k-1}, r_{k-1})$       ▷ Select an augmentation according to the recurrent policy
7:      Update the policy and value function according to Algorithm 1 with $f = f_k$ and $K = 1$:
8:      $\theta \leftarrow \arg\max_\theta J_{\text{DrAC}}$       ▷ Update the policy
9:      $\phi \leftarrow \arg\max_\phi J_{\text{DrAC}}$       ▷ Update the value function
10:      Compute the mean return obtained by the new policy $r_k$.
11:      Reward $g_\psi$ with $r_k$.
12:      Update $g_\psi$ using REINFORCE.
13: **end for**

---

---

**Algorithm 4 Meta-DrAC**

---

1: **Hyperparameters:** Distribution over tasks (or levels) $q(m)$, number of updates K, step size parameters $\alpha$ and $\beta$, initial policy $\pi_\theta$, initial value function $V_\phi$.
2: Initialize the set of all training levels $\mathcal{D} = \{l\}_{i=1}^L$.
3: Initialize the augmentation as a CNN $g$ with parameters $\psi$.
4: **for** $k = 1, \ldots, K$ **do** a batch of tasks $m_i \sim q(m)$
5:      **for all** $m_i$ **do**
6:          Collect trajectories on task $m_i$ using the current policy.
7:          Update the policy and value function according to Algorithm 1 with $f = f_k$ and $K = 1$:
8:          $\theta \leftarrow \arg\max_\theta J_{\text{DrAC}}$       ▷ Update the policy
9:          $\phi \leftarrow \arg\max_\phi J_{\text{DrAC}}$       ▷ Update the value function
10:          Compute the return of the new agent on task $m_i$ after being updated with $g_\psi$, $r_{m_i}(g_\psi)$
11:          $\psi_i' \leftarrow \psi + \alpha \nabla_\psi r_{m_i}(g_\psi)$
12:      **end for**
13:      $\psi \leftarrow \psi + \beta \nabla_\psi \sum_{m_i \sim q(m)} r_{m_i}(g_{\psi_i'})$
14: **end for**

---

Table 3: List of hyperparameters used to obtain the results in this paper.

| Hyperparameter | Value |
|---|---|
| $\gamma$ | 0.999 |
| $\lambda$ | 0.95 |
| # timesteps per rollout | 256 |
| # epochs per rollout | 3 |
| # minibatches per epoch | 8 |
| entropy bonus | 0.01 |
| clip range | 0.2 |
| reward normalization | yes |
| learning rate | 5e-4 |
| # workers | 1 |
| # environments per worker | 64 |
| # total timesteps | 25M |
| optimizer | Adam |
| LSTM | no |
| frame stack | no |
| $\alpha_r$ | 0.1 |
| c | 0.1 |
| K | 10 |

## E HYPERPARAMETERS

We use Kostrikov (2018)'s implementation of PPO (Schulman et al., 2017), on top of which all our methods are build. The agent is parameterized by the ResNet architecture from (Espeholt et al., 2018) which was used to obtain the best results in Cobbe et al. (2019). Following Cobbe et al. (2019), we also share parameters between the policy an value networks. To improve stability when training with DrAC, we only backpropagate gradients through $\pi(a|f(s,\nu))$ and $V(f(s,\nu))$ in equations (3) and (4), respectively. Unless otherwise noted, we use the best hyperparameters found in Cobbe et al. (2019) for the easy mode of Procgen (*i.e.* same experimental setup as the one used here), namely:

For DrAC, we did a grid search for the regularization coefficient $\alpha_r \in [0.0001, 0.01, 0.05, 0.1, 0.5, 1.0]$ used in equation (5) and found that the best value is $\alpha_r = 0.1$, which was used to produce all the results in this paper.

For UCB-DrAC, we did grid searches for the exploration coefficient $c \in [0.0, 0.1, 0.5, 1.0, 5.0]$ and the size of the sliding window used to compute the Q-values $K \in [10, 50, 100]$. We found that the best values are $c = 0.1$ and $K = 10$, which were used to obtain the results shown here.

For RL2-DrAC, we performed a hyperparameter search for the dimension of recurrent hidden state $h \in [16, 32, 64]$, for the learning rate $l \in [3e-4, 5e-4, 7e-4]$, and for the entropy coefficient $e \in [1e-4, 1e-3, 1e-2]$ and found $h = 32$, $l = 5e-4$, and $e = 1e-3$ to work best. We used Adam with $\epsilon = 1e-5$ as the optimizer.

For Meta-DrAC, the convolutional network whose weights we meta-learn consists of a single convolutional layer with 3 input and 3 output channels, kernel size 3, stride 1 and 0 padding. At each epoch, we perform one meta-update where we unroll the inner optimizer using the training set and compute the meta-test return on the validation set. We did the same hyperparameter grid searches as for RL2-DrAC and found that the best values were $l = 7e-4$ and $e = 1e-2$ in this case. The buffer of experience (collected before each PPO update) was split into $90\%$ for meta-training and $10\%$ for meta-testing.

For Rand-FM Lee et al. (2020) we use the recommended hyperparameters in the authors' released implementation, which were the best values for CoinRun (Cobbe et al., 2018), one of the Procgen games used for evaluation in (Lee et al., 2020).

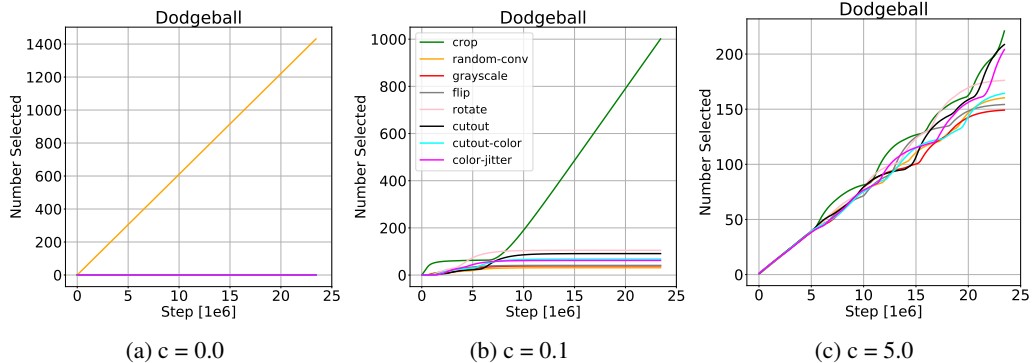

Figure 5: Behavior of UCB for different values of its exploration coefficient c on Dodgeball. When c is too small, UCB might converge to a suboptimal augmentation. On the other hand, when c is too large, UCB might take too long to converge.

For IBAC-SNI Igl et al. (2019) we also use the authors' open sourced implementation. We use the parameters corresponding to IBAC-SNI $\lambda = .5$. We use weight regularization with $l_2 = .0001$, data augmentation turned on, and a value of $\beta = .0001$ which turns on the variational information bottleneck, and selective noise injection turned on. This corresponds to the best version of this approach, as found by the authors after evaluating it on CoinRun (Cobbe et al., 2018). While IBAC-SNI outperforms the other methods on maze-like games such as heist, maze, and miner, it is still significantly worse than our approach on the entire Procgen benchmark.

For both baselines, Rand-FM and IBAC-SNI, we use the same experimental setup for training and testing as the one used for our methods. Hence, we train them for 25M frames on the easy mode of each Procgen game, using (the same) 200 levels for training and the rest of the levels for testing.

We use the Adam (Kingma & Ba, 2015) optimizer for all our experiments. Note that by using Adam, we do not need separate coefficients for the policy and value regularization terms (since Adam rescales gradients for each loss component accordingly).

## F  ANALYSIS OF UCB'S BEHAVIOR

In Figure 3, we show the behavior of UCB during training, along with train and test performance on the respective environments. In the case of Ninja, UCB converges to always selecting the best augmentation only after 15M training steps. This is because the augmentations have similar effects on the agent early in training, so it takes longer to find the best augmentation from the given set. In contrast, on Dodgeball, UCB finds the most effective augmentation much earlier in training because there is a significant difference between the effect of various augmentations. Early discovery of an effective augmentation leads to significant improvements over PPO, for both train and test environments.

Another important factor is the exploration coefficient used by UCB (see equation (6)) to balance the exploration and exploitation of different augmentations. Figure 5 compares UCB's behavior for different values of the exploration coefficient. Note that if the coefficient is 0, UCB always selects the augmentation with the largest Q-value. This can sometimes lead to UCB converging on a suboptimal augmentation due to the lack of exploration. However, if the exploration term of equation (6) is too large relative to the differences in the Q-values among various augmentations, UCB might take too long to converge. In our experiments, we found that an exploration coefficient of 0.1 results in a good exploration-exploitation balance and works well across all Procgen games.

# G  PROCGEN BENCHMARK

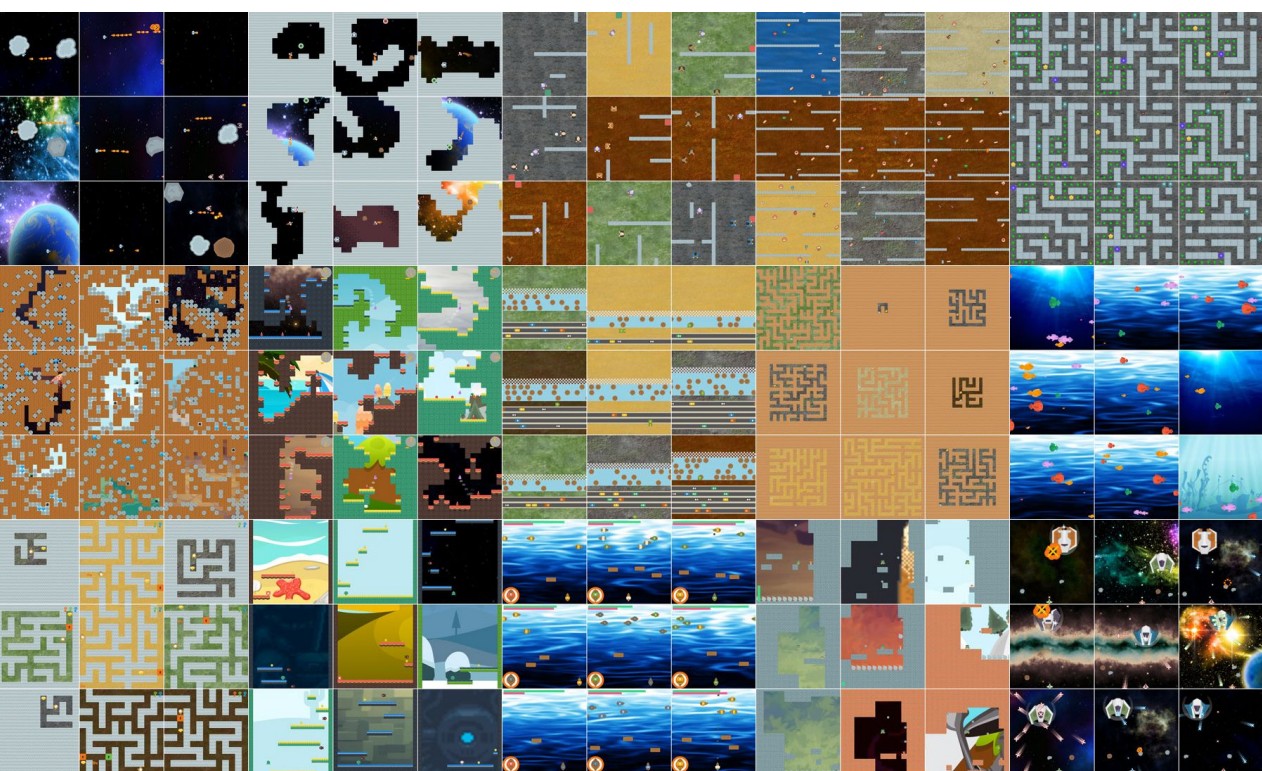

Figure 6: Screenshots of multiple procedurally-generated levels from 15 Procgen environments: StarPilot, CaveFlyer, Dodgeball, FruitBot, Chaser, Miner, Jumper, Leaper, Maze, BigFish, Heist, Climber, Plunder, Ninja, BossFight (from left to right, top to bottom).

# H  BEST AUGMENTATIONS

Table 4: Best augmentation type for each game, as evaluated on the test environments.

| Game | BigFish | StarPilot | FruitBot | BossFight | Ninja | Plunder | CaveFlyer | CoinRun |
|---|---|---|---|---|---|---|---|---|
| Best Augmentation | crop | crop | crop | flip | color-jitter | crop | rotate | random-conv |

Table 5: Best augmentation type for each game, as evaluated on the test environments.

| Game | Jumper | Chaser | Climber | Dodgeball | Heist | Leaper | Maze | Miner |
|---|---|---|---|---|---|---|---|---|
| Best Augmentation | random-conv | crop | color-jitter | crop | crop | crop | crop | color-jitter |

# I BREAKDOWN OF PROCGEN SCORES

Table 6: Procgen scores on train levels after training on 25M environment steps. The mean and standard deviation are computed using 10 runs. The best augmentation for each game is used when computing the results for DrAC and RAD.

| Game | PPO | Rand + FM | IBAC-SNI | DrAC | RAD | UCB-DrAC | RL2-DrAC | Meta-DrAC |
|---|---|---|---|---|---|---|---|---|
| BigFish | $8.9 \pm 1.5$ | $6.0 \pm 0.9$ | $\mathbf{19.1 \pm 0.8}$ | $13.1 \pm 2.2$ | $13.2 \pm 2.8$ | $13.2 \pm 2.2$ | $10.1 \pm 1.9$ | $9.28 \pm 1.9$ |
| StarPilot | $29.8 \pm 2.3$ | $26.3 \pm 0.8$ | $26.7 \pm 0.7$ | $\mathbf{38.0 \pm 3.1}$ | $36.5 \pm 3.9$ | $35.3 \pm 2.2$ | $30.6 \pm 2.6$ | $30.5 \pm 3.9$ |
| FruitBot | $29.1 \pm 1.1$ | $29.2 \pm 0.7$ | $29.4 \pm 0.8$ | $29.4 \pm 1.0$ | $26.1 \pm 3.0$ | $\mathbf{29.5 \pm 1.2}$ | $29.2 \pm 1.0$ | $29.4 \pm 1.1$ |
| BossFight | $\mathbf{8.5 \pm 0.7}$ | $5.6 \pm 0.7$ | $7.9 \pm 0.7$ | $8.2 \pm 1.0$ | $8.1 \pm 1.1$ | $8.2 \pm 0.8$ | $8.4 \pm 0.8$ | $7.9 \pm 0.5$ |
| Ninja | $7.4 \pm 0.7$ | $7.2 \pm 0.6$ | $8.3 \pm 0.8$ | $8.8 \pm 0.5$ | $\mathbf{8.9 \pm 0.9}$ | $8.5 \pm 0.3$ | $8.1 \pm 0.6$ | $7.8 \pm 0.4$ |
| Plunder | $6.0 \pm 0.5$ | $5.5 \pm 0.7$ | $6.0 \pm 0.6$ | $9.9 \pm 1.3$ | $8.4 \pm 1.5$ | $\mathbf{11.1 \pm 1.6}$ | $5.3 \pm 0.5$ | $6.5 \pm 0.5$ |
| CaveFlyer | $6.8 \pm 0.6$ | $6.5 \pm 0.5$ | $6.2 \pm 0.5$ | $\mathbf{8.2 \pm 0.7}$ | $6.0 \pm 0.8$ | $5.7 \pm 0.6$ | $5.3 \pm 0.8$ | $6.5 \pm 0.7$ |
| CoinRun | $9.3 \pm 0.3$ | $9.6 \pm 0.6$ | $9.6 \pm 0.4$ | $\mathbf{9.7 \pm 0.2}$ | $9.6 \pm 0.4$ | $9.5 \pm 0.3$ | $9.1 \pm 0.3$ | $9.4 \pm 0.2$ |
| Jumper | $8.3 \pm 0.4$ | $8.9 \pm 0.4$ | $8.5 \pm 0.6$ | $\mathbf{9.1 \pm 0.4}$ | $8.6 \pm 0.4$ | $8.1 \pm 0.7$ | $8.6 \pm 0.4$ | $8.4 \pm 0.5$ |
| Chaser | $4.9 \pm 0.5$ | $2.8 \pm 0.7$ | $3.1 \pm 0.8$ | $\mathbf{7.1 \pm 0.5}$ | $6.4 \pm 1.0$ | $7.6 \pm 1.0$ | $4.5 \pm 0.7$ | $5.5 \pm 0.8$ |
| Climber | $8.4 \pm 0.8$ | $7.5 \pm 0.8$ | $7.1 \pm 0.7$ | $\mathbf{9.9 \pm 0.8}$ | $9.3 \pm 1.1$ | $9.0 \pm 0.4$ | $7.9 \pm 0.9$ | $8.5 \pm 0.5$ |
| Dodgeball | $4.2 \pm 0.5$ | $4.3 \pm 0.3$ | $\mathbf{9.4 \pm 0.6}$ | $7.5 \pm 1.0$ | $5.0 \pm 0.7$ | $8.3 \pm 0.9$ | $6.3 \pm 1.1$ | $4.8 \pm 0.6$ |
| Heist | $\mathbf{7.1 \pm 0.5}$ | $6.0 \pm 0.5$ | $4.8 \pm 0.7$ | $6.8 \pm 0.7$ | $6.2 \pm 0.9$ | $6.9 \pm 0.4$ | $5.6 \pm 0.8$ | $6.6 \pm 0.6$ |
| Leaper | $\mathbf{5.5 \pm 0.4}$ | $3.2 \pm 0.7$ | $2.7 \pm 0.4$ | $5.0 \pm 0.7$ | $4.9 \pm 0.9$ | $5.3 \pm 0.5$ | $2.7 \pm 0.6$ | $3.7 \pm 0.6$ |
| Maze | $9.1 \pm 0.3$ | $8.9 \pm 0.6$ | $8.2 \pm 0.8$ | $8.3 \pm 0.7$ | $8.4 \pm 0.7$ | $8.7 \pm 0.6$ | $7.0 \pm 0.7$ | $\mathbf{9.2 \pm 0.2}$ |
| Miner | $12.2 \pm 0.3$ | $11.7 \pm 0.8$ | $8.5 \pm 0.7$ | $12.5 \pm 0.3$ | $\mathbf{12.6 \pm 1.0}$ | $12.5 \pm 0.2$ | $10.9 \pm 0.5$ | $12.4 \pm 0.3$ |

Table 7: Procgen scores on test levels after training on 25M environment steps. The mean and standard deviation are computed using 10 runs. The best augmentation for each game is used when computing the results for DrAC and RAD.

| Game | PPO | Rand + FM | IBAC-SNI | DrAC | RAD | UCB-DrAC | RL2-DrAC | Meta-DrAC |
|---|---|---|---|---|---|---|---|---|
| BigFish | $4.0 \pm 1.2$ | $0.6 \pm 0.8$ | $0.8 \pm 0.9$ | $8.7 \pm 1.4$ | $\mathbf{9.9 \pm 1.7}$ | $9.7 \pm 1.0$ | $6.0 \pm 0.5$ | $3.3 \pm 0.6$ |
| StarPilot | $24.7 \pm 3.4$ | $8.8 \pm 0.7$ | $4.9 \pm 0.8$ | $29.5 \pm 5.4$ | $\mathbf{33.4 \pm 5.1}$ | $30.2 \pm 2.8$ | $29.4 \pm 2.0$ | $26.6 \pm 2.8$ |
| FruitBot | $26.7 \pm 0.8$ | $24.5 \pm 0.7$ | $24.7 \pm 0.8$ | $28.2 \pm 0.8$ | $27.3 \pm 1.8$ | $\mathbf{28.3 \pm 0.9}$ | $27.5 \pm 1.6$ | $27.4 \pm 0.8$ |
| BossFight | $7.7 \pm 1.0$ | $1.7 \pm 0.9$ | $1.0 \pm 0.7$ | $7.5 \pm 0.8$ | $7.9 \pm 0.6$ | $\mathbf{8.3 \pm 0.8}$ | $7.6 \pm 0.9$ | $7.7 \pm 0.7$ |
| Ninja | $5.9 \pm 0.7$ | $6.1 \pm 0.8$ | $\mathbf{9.2 \pm 0.6}$ | $7.0 \pm 0.4$ | $6.9 \pm 0.8$ | $6.9 \pm 0.6$ | $6.2 \pm 0.5$ | $5.9 \pm 0.7$ |
| Plunder | $5.0 \pm 0.5$ | $3.0 \pm 0.6$ | $2.1 \pm 0.8$ | $\mathbf{9.5 \pm 1.0}$ | $8.5 \pm 1.2$ | $8.9 \pm 1.0$ | $4.6 \pm 0.3$ | $5.6 \pm 0.4$ |
| CaveFlyer | $5.1 \pm 0.9$ | $5.4 \pm 0.8$ | $\mathbf{8.0 \pm 0.8}$ | $6.3 \pm 0.8$ | $5.1 \pm 0.6$ | $5.3 \pm 0.9$ | $4.1 \pm 0.9$ | $5.5 \pm 0.4$ |
| CoinRun | $8.5 \pm 0.5$ | $\mathbf{9.3 \pm 0.4}$ | $8.7 \pm 0.6$ | $8.8 \pm 0.$ | $9.0 \pm 0.8$ | $8.5 \pm 0.6$ | $8.3 \pm 0.5$ | $8.6 \pm 0.5$ |
| Jumper | $5.8 \pm 0.5$ | $5.3 \pm 0.6$ | $3.6 \pm 0.6$ | $\mathbf{6.6 \pm 0.4}$ | $6.5 \pm 0.6$ | $6.4 \pm 0.6$ | $6.5 \pm 0.5$ | $5.8 \pm 0.7$ |
| Chaser | $5.0 \pm 0.8$ | $1.4 \pm 0.7$ | $1.3 \pm 0.5$ | $5.7 \pm 0.6$ | $5.9 \pm 1.0$ | $\mathbf{6.7 \pm 0.6}$ | $3.8 \pm 0.5$ | $5.1 \pm 0.6$ |
| Climber | $5.7 \pm 0.8$ | $5.3 \pm 0.7$ | $3.3 \pm 0.6$ | $\mathbf{7.1 \pm 0.7}$ | $6.9 \pm 0.8$ | $6.5 \pm 0.8$ | $6.3 \pm 0.5$ | $6.6 \pm 0.6$ |
| Dodgeball | $\mathbf{11.7 \pm 0.3}$ | $0.5 \pm 0.4$ | $1.4 \pm 0.4$ | $4.3 \pm 0.8$ | $2.8 \pm 0.7$ | $4.7 \pm 0.7$ | $3.0 \pm 0.8$ | $1.9 \pm 0.5$ |
| Heist | $2.4 \pm 0.5$ | $2.4 \pm 0.6$ | $\mathbf{9.8 \pm 0.6}$ | $4.0 \pm 0.8$ | $4.1 \pm 1.0$ | $4.0 \pm 0.7$ | $2.4 \pm 0.4$ | $2.0 \pm 0.6$ |
| Leaper | $4.9 \pm 0.7$ | $6.2 \pm 0.5$ | $\mathbf{6.8 \pm 0.6}$ | $5.3 \pm 1.1$ | $4.3 \pm 1.0$ | $5.0 \pm 0.3$ | $2.8 \pm 0.7$ | $3.3 \pm 0.4$ |
| Maze | $5.7 \pm 0.6$ | $8.0 \pm 0.7$ | $\mathbf{10.0 \pm 0.7}$ | $6.6 \pm 0.8$ | $6.1 \pm 1.0$ | $6.3 \pm 0.6$ | $5.6 \pm 0.3$ | $5.2 \pm 0.6$ |
| Miner | $8.5 \pm 0.5$ | $7.7 \pm 0.6$ | $8.0 \pm 0.6$ | $\mathbf{9.8 \pm 0.6}$ | $9.4 \pm 1.2$ | $9.7 \pm 0.7$ | $8.0 \pm 0.4$ | $9.2 \pm 0.7$ |

## J    PROCGEN LEARNING CURVES

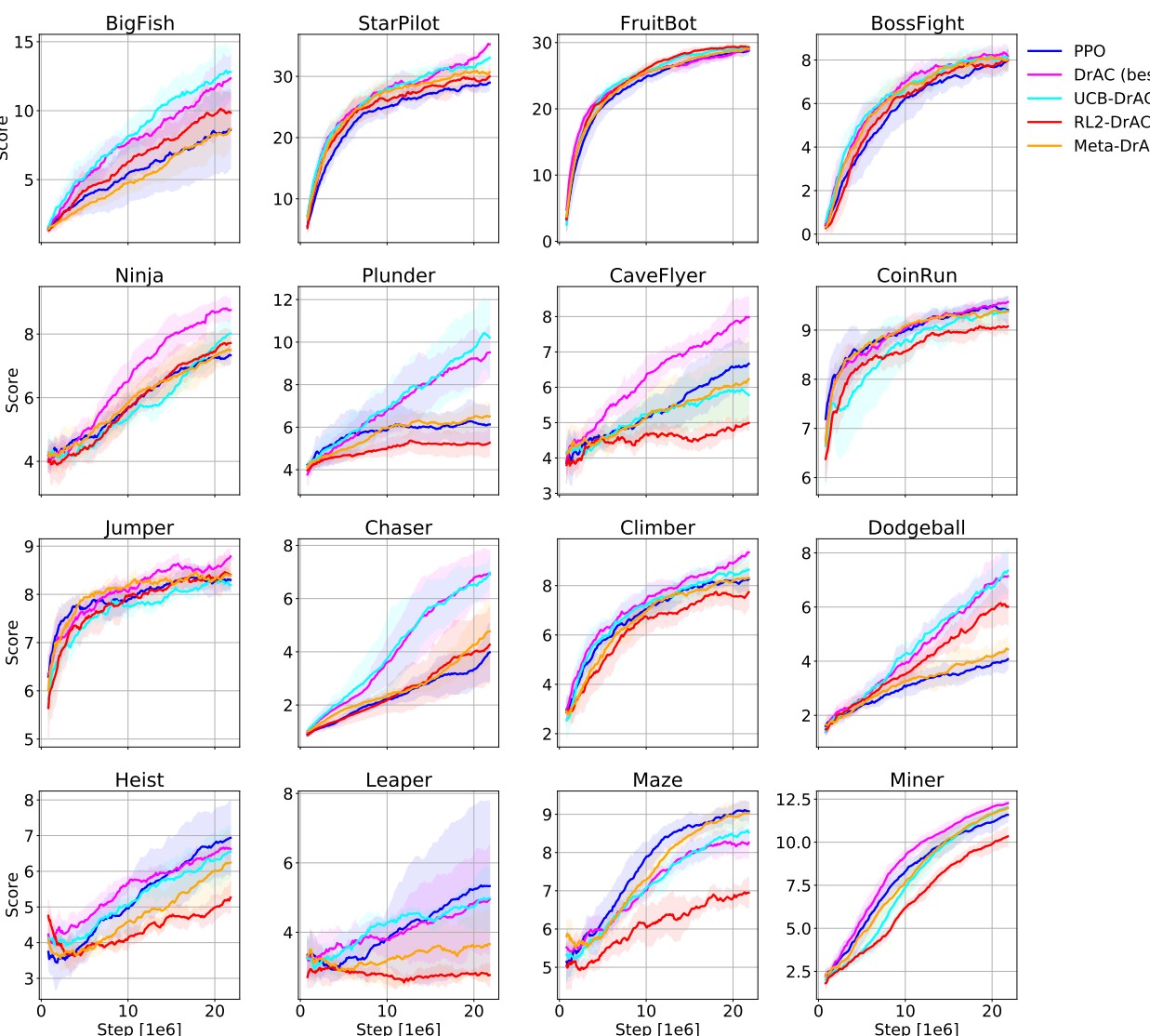

Figure 7: Train performance of various approaches that automatically select an augmentation, namely UCB-DrAC, RL2-DrAC, and Meta-DrAC. The mean and standard deviation are computed using 10 runs.

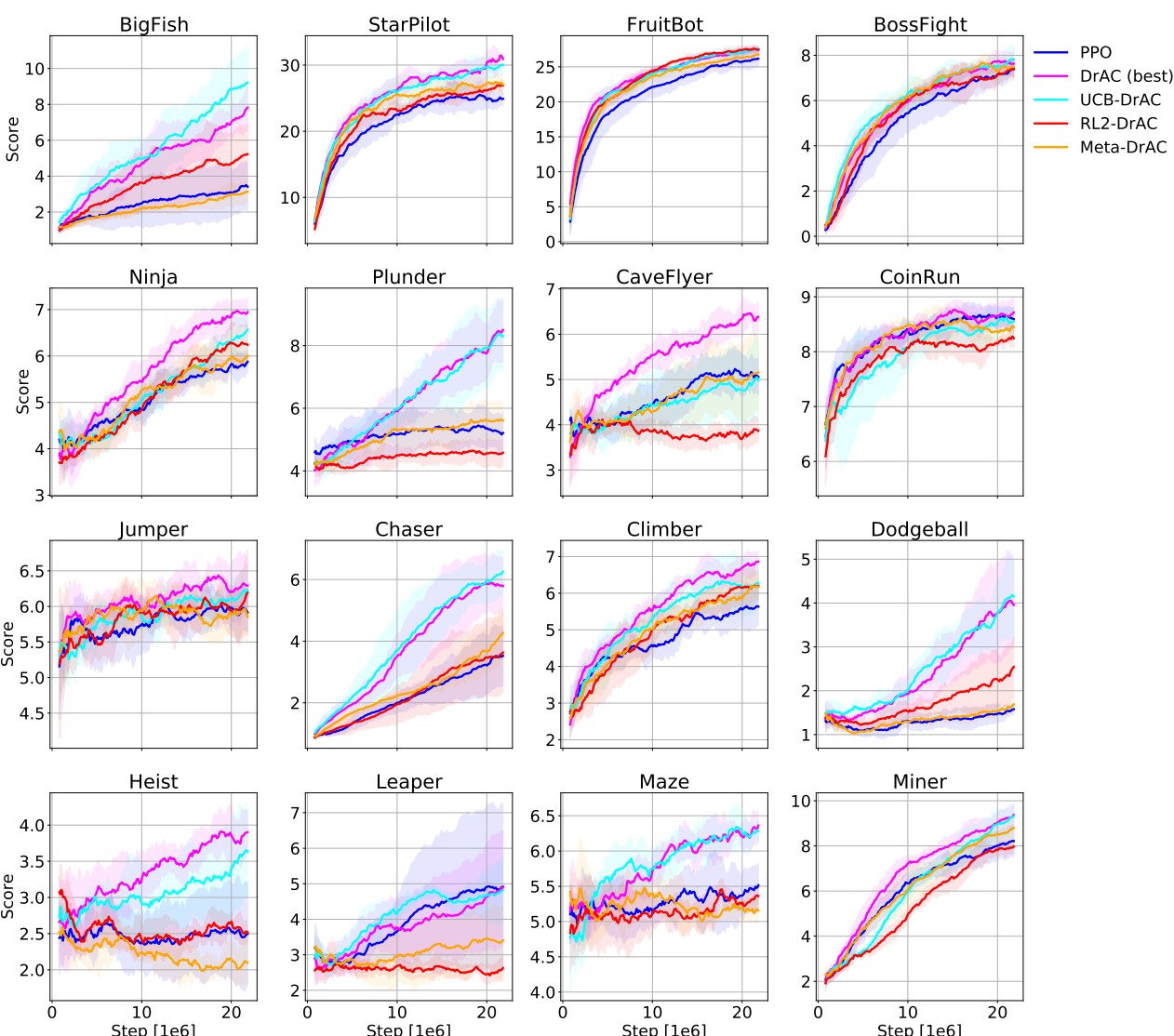

Figure 8: Test performance of various approaches that automatically select an augmentation, namely UCB-DrAC, RL2-DrAC, and Meta-DrAC. The mean and standard deviation are computed using 10 runs.

# K DEEPMIND CONTROL SUITE EXPERIMENTS

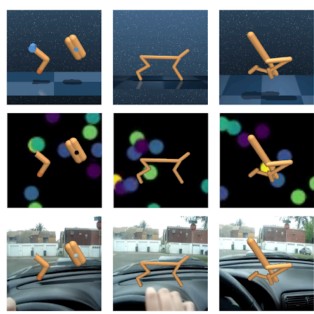

Figure 9: DMC environment examples. Top row: default backgrounds without any distractors. Middle row: simple distractor backgrounds with ideal gas videos. Bottom row: natural distractor backgrounds with Kinetics videos. Tasks from left to right: Finger Spin, Cheetah Run, Walker Walk.

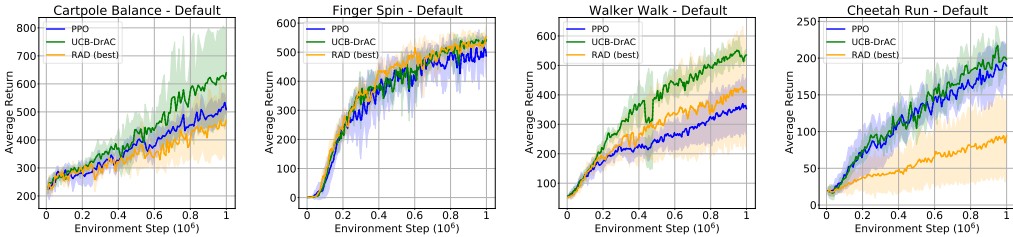

Figure 10: Average return on DMC tasks with default (*i.e.* no distractor) backgrounds with mean and standard deviation computed over 5 seeds. UCB-DrAC outperforms PPO and RAD with the best augmentations.

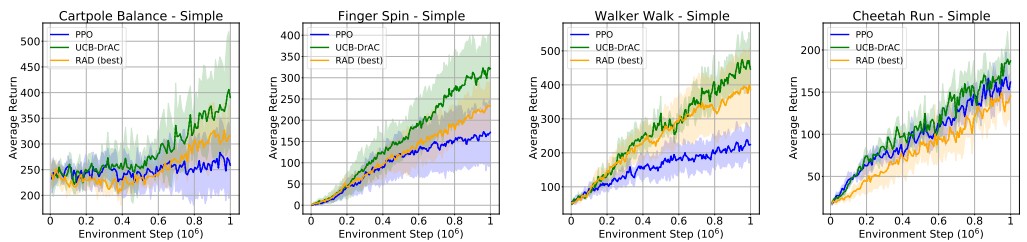

Figure 11: Average return on DMC tasks with simple (*i.e.* synthetic) distractor backgrounds with mean and standard deviation computed over 5 seeds. UCB-DrAC outperforms PPO and RAD with the best augmentations.

For our DMC experiments, we ran a grid search over the learning rate in $[1e-4, 3e-4, 7e-4, 1e-3]$, the number of minibatches in $[32, 8, 16, 64]$, the entropy coefficient in $[0.0, 0.01, 0.001, 0.0001]$, and the number of ppo epochs per update in $[3, 5, 10, 20]$. For Walker Walk and Finger Spin we use 2 action repeats and for the others we use 4. We also use 3 stacked frames as observations. For Finger Spin, we found 10 ppo epochs, 0.0 entropy coefficient, 16 minibatches, and 0.0001 learning rate to be the best. For Cartpole Balance, we used the same except for 0.001 learning rate. For Walker Walk, we used 5 ppo epochs, 0.001 entropy coefficient, 32 minibatches, and 0.001 learning rate. For Cheetah Run, we used 3 ppo epochs, 0.0001 entropy coefficient, 64 minibatches, and 0.001 learning rate. We use $\gamma = 0.99$, $\gamma = 0.95$ for the generalized advantage estimates, 2-48 steps, 1 process, value loss coefficient 0.5, and linear rate decay over 1 million environment steps. We also found crop to be the best augmentation from our set of eight transformations. Any other hyperparameters not mentioned here were set to the same values as the ones used for Procgen as described above.

