# OpenReview forum: "Automatic Data Augmentation for Generalization in Reinforcement Learning"
_ICLR.cc/2021/Conference — Reject_

### Official Review · AnonReviewer3 · 2020-10-19
**Tend to accept**

**Rating:** 6
**Confidence:** 3

**Review:**

Summary:

This paper proposes an automatic data augmentation approach for RL tasks. Specifically, it takes UCB for data selection and introduces two regularization terms for actor-critic algorithms' policy and value function. Then this paper evaluated the approach based on the Procgen benchmark and demonstrated that it outperforms existing methods. It is also shown that the learned policies and representations are robust to irrelevant factors.


Reasons for score:

On the one hand, I feel that the proposed approach is incremental (UCB for data selection with the actor-critic algorithm as the RL algorithm plus additional regularization terms). It would also be ideal if more experiments can be included to prove that the proposed approach is effective in most RL tasks. However, on the other hand, based on the current experimentation results, the proposed method seems promising and can be useful for generalization in reinforcement learning.


Pros

1. The proposal of the automatic data augmentation/selection approach is useful, given that the existing primary methods either rely on expert knowledge or separately evaluate a large number of transformations to find the best one, both of which are expensive.

2. The proposed approach achieves SOTA performance in the Procgen benchmark.

3. It shows that the proposed method is robust to irrelevant factors.


Cons

1. I think the primary difference between the proposed approach and existing approaches is the additional regularization losses. This would make the new approach seem incremental.

2. There is a lack of performance comparison between the automatic data augmentation method and non-automatic approaches.

3. The data augmentation approach is only evaluated in the Procgen benchmark. It would be great if this paper includes more experiments to demonstrate the performance, especially since the proposed method is intended for any RL task.

---

> ### Author Response · Authors · 2020-11-12
> **Thank you for your feedback!**
>
> We thank their reviewer for their positive and constructive comments. We particularly appreciate that the reviewer finds our approach to be useful, effective and robust to irrelevant factors. We respond to the reviewer’s comments below.
>
> C1: “I think the primary difference between the proposed approach and existing approaches is the additional regularization losses. This would make the new approach seem incremental.”
>
> A1: We respectfully disagree with the characterization that our approach is “incremental”. On the contrary, we believe that the simplicity of our approach is one of its main strengths since it makes it easier build upon for other members of the community. In addition, even if our method is simple, it shows significant improvements over the previous state-of-the-art on a challenging benchmark for generalization in RL. Moreover, we provide theoretical intuition and empirical evidence for the importance of the additional regularization losses we introduce. As shown in Figure 2, not using these additional losses, the methods may completely fail to learn in certain cases.
>
> C2: “There is a lack of performance comparison between the automatic data augmentation method and non-automatic approaches.”
>
> A2: Figure 6 and 7 in the Appendix compare our proposed automatic approaches (UCB-DrAC, RL2-DrAC, and Meta-DrAC) with the non-automatic ones (DrAC with the best augmentation).
>
> C3: “The data augmentation approach is only evaluated in the Procgen benchmark. It would be great if this paper includes more experiments to demonstrate the performance, especially since the proposed method is intended for any RL task.”
>
> A3: We agree with the reviewer that it would be valuable to include experiments on other domains and we will do our best to report back on this by the end of the discussion period. However, we would like to point out that our paper already contains extensive empirical results, with 25 models on 16 different environments, 10 seeds each, accounting to 4000 experiments in total, each taking at least 4 hours on a GPU (after hyperparameter search). Please note that this is far in excess of most other published papers that use Procgen for evaluation, which showed results only on a subset of the environments (e.g. 3 in Laskin et al. 2020, 1 in Igl et al. 2019 and Lee et al. 2020). While we aim to include experiments on another domain by the end of the discussion period, due to computational and time constraints, we expect the comparisons to be more limited in scope (i.e. they will likely contain a smaller number of environments and baselines).
>
> In light of these clarifications, we would appreciate it if the reviewer would be willing to reconsider their assessment and/or provide us with further feedback.

---

### Official Review · AnonReviewer2 · 2020-10-25
**An effective framework for automatic data augmentation in deep RL**

**Rating:** 7
**Confidence:** 3

**Review:**

#######################################

Summary:
This paper tackles the problem of generalization in deep RL via data augmentation. It provides a framework for automatic data augmentation based on UCB, RL^2, or MAML. When UCB is combined with regularization of the policy and value function so that their outputs are invariant to transformations (such as rotation, crop, etc.), it shows improvements over similar algorithms on the ProcGen benchmark.

#######################################

Pros:
1. The paper is well written and the proposed algorithms are straightforward to understand.
2. UCB-DrAC is shown to be statistically superior to the tested baselines. Surprisingly, when all ProcGen games are taken into account, some existing methods such as Rand-FM are actually worse than PPO.
3. Ablation studies are able to show that both ingredients of UCB-DrAC are important. UCB is shown to find the best augmentation asymptotically, and the DrAC is shown to be better than PPO and RAD.

Cons:
1. The experiments do not compare the proposed algorithms to DrQ, the algorithm proposed in Kostrikov et al. (2020). Since it also tackles generalization in deep RL through data augmentation, it seems that it should be included as a baseline. Can the authors explain why it was not included?
2. The proposed algorithms are only combined with PPO. It would be good to have some results for other actor-critic algorithms, such as SAC, to verify if the SOTA behavior holds.

#######################################

Overall:
I would vote for accepting this paper, due to the strength of its experimental results. The proposed approaches are novel in the data augmentation for generalization in deep RL subfield, although AutoAugment (cited in the paper) also uses RL for choosing data augmentations in the supervised learning case.

#######################################

Further comments and questions:
1. It may be helpful to rewrite some of the algorithms in Section 3.3 in pseudocode, as text is harder to read.
2. Is there intuition for why only UCB-DrAC leads to statistically significant improvements over PPO, and not RL2-DrAC and Meta-DrAC? Can it be shown that the former has lower sample complexity than the latter two?
3. Why is the cycle consistency percentage in Table 2 always low? Is it some idiosyncrasy of the metric or due to the inherent variance in the trajectories?

#######################################

Update after reading other reviews and author responses:

I am happy to keep my score and support accepting this paper. I agree with Reviewer 4 that conducting a more detailed ablation study of the impact of regularizing both the policy and value function (i.e. a comparison with an algorithm like that of Kostrikov et al. (2020)) would improve the paper and hope that it will be included in the final paper.

---

> ### Author Response · Authors · 2020-11-12
> **Thank you for your feedback!**
>
> We thank the reviewer for their positive and constructive feedback. We were delighted to hear they found our paper to be well written, our experimental results strong, and our approach novel for generalization in RL. We address your concerns below.
>
> C1: “The experiments do not compare the proposed algorithms to DrQ, the algorithm proposed in Kostrikov et al. (2020). Since it also tackles generalization in deep RL through data augmentation, it seems that it should be included as a baseline. Can the authors explain why it was not included?”
>
> A1: The reason we do not compare with DrQ is because DrQ uses SAC as a base algorithm, while our method is based on PPO. We chose to use PPO as a base algorithm because it is the state-of-the-art on the Procgen benchmark, which we use to test generalization. In addition, TD(0) methods such as SAC or DrQ have not been shown yet to achieve decent performance on Procgen. Please also note that DrQ was designed for improving sample complexity rather than generalization.
>
> C2: “The proposed algorithms are only combined with PPO. It would be good to have some results for other actor-critic algorithms, such as SAC, to verify if the SOTA behavior holds.”
>
> A2: As mentioned above, to the best of our knowledge SAC is not well suited for the Procgen benchmark. While we are happy to run more experiments with other actor-critic algorithms, we would like to point out that our paper already contains extensive empirical results, with 25 models on 16 different environments, 10 seeds each, accounting to 4000 experiments in total, each taking at least 4 hours on a GPU (after hyperparameter search). Please note that this is far in excess of most other published papers that use Procgen for evaluation, which showed results only on a subset of the environments (e.g. 3 in Laskin et al. 2020, 1 in Igl et al. 2019 and Lee et al. 2020).
>
> C3: “It may be helpful to rewrite some of the algorithms in Section 3.3 in pseudocode, as text is harder to read.”
>
> A3: Thank you for the suggestion. We will add a pseudocode in the paper.
>
> C4: “Is there intuition for why only UCB-DrAC leads to statistically significant improvements over PPO, and not RL2-DrAC and Meta-DrAC? Can it be shown that the former has lower sample complexity than the latter two?”
>
> A4: We are not entirely sure but UCB does have theoretical finite-time regret guarantees for multi-armed bandit problems such as this one (Auer, 2002) while we are not aware of a comparable result for RL^2 or MAML. We also think it is possible that UCB is better suited for our problem formulation while RL^2 and MAML start showing their benefits on more complex problems such as contextual bandits or sequential decision making. We think this is an interesting question for future work.
>
> C5: "Why is the cycle consistency percentage in Table 2 always low? Is it some idiosyncrasy of the metric or due to the inherent variance in the trajectories?"
>
> A5: We believe this might indeed be due to the variance in the trajectories. In some cases, there can be more than a single path for maximizing reward and the agent might be prone to choose different paths for different backgrounds of the same level (which could be exacerbated by the fact that many of the backgrounds are highly structured rather uniform).
>
> In light of these clarifications, we would appreciate it if the reviewer confirmed that all their concerns have been addressed and, if so, reconsider their assessment.

---

### Official Review · AnonReviewer4 · 2020-10-25
**Data sugmentation is a nice tool for RL, the authors propose a simple and effective solution but fail to motivate their method or support theoretical claims**

**Rating:** 4
**Confidence:** 4

**Review:**

Summary after Discussion Period:
-----------------------------------------------
After reading the other reviewer's comments and corresponding with the authors, I have become convinced that the author's proposed regularization method is novel and effective, and would recommend this avenue of research be further explored. Yet it has also become clear to me that the author's claims on why their method works are not yet supported by evidence. Further, I don't believe the author's proposed further ablation studies would fix the theory, since such experiments don't address whether their method works by fixing problems with Laskin’s work (as the author's claim) or because it provides a more direct way of enforcing invariance to transformation (as I claim).

So we're left with a difficult situation, the method and the experiments are good while the theory is lacking. In such a situation both acceptance or rejection seem reasonable. Yet, as per ICLR reviewer guidelines, one should answer three questions for oneself:

 - What is the specific question and/or problem tackled by the paper?
 - is the approach well motivated, including being well-placed in the literature?
 - Does the paper support the claims? This includes determining if results, whether theoretical or empirical, are correct and if they are scientifically rigorous.

Since the theory is lacking and the approach is not well motivated, and since the theoretical claims haven't been rigorously supported, I feel as per ICLR guidelines the paper is not yet ready for acceptance.

Initial Review:
-------------------

Summary In this paper, the authors introduce three approaches for automatically finding an augmentations for any RL task, and a novel regularization scheme to make such augmentations work effectively.

Positive aspects:
-----------------------
The paper’s language is clear and the authors provide a good overview of the problem of data augmentation for reinforcement learning. Furthermore, they nicely explain why data augmentation for RL isn’t as straightforward as augmenting data for supervised learning learning. I believe that data augmentation could be a nice tool in the Reinforcement Learner’s toolbox, and I’m glad to see a paper advancing the idea.


Major Concerns:
-----------------------

This paper has not provided sufficient evidence that the author’s proposed way of doing data augmentation is effective. In this paper, there are two main novel methods for doing data augmentation / insights in RL. I will discuss my concerns with both methods separately.

Policy and Value function Regularization.
-----------------------------------------------------

The authors criticize the naive application of transformers in the PPO’s buffer, as done in Laskin et al. (2020), saying that this changes the PPO objective. While I agree that such a naive transformation as in Eq. 2 is problematic, I fail to understand why application of transformation to states in the buffer would result in the Eq. 2, as the transformation would happen to the states being fed into both $\pi_\theta$ and $\pi_{\theta_\text{old}}$, resulting in an equation different from Eq. 2. One just needs to save the old policies (and old Advantage function), so that $\pi_{\theta_\text{old}}$ can be applied to transformed states, and not just use the actions from the buffer. This would seem like a straightforward fix.

Yet the authors have proposed a different regularization fix which judging the experiments does seem to work, as shown in Figure 2. I suspect it works for another reason: since the regularization forces $V(s) = V(f(s))$ and $\pi(\cdot | s) = \pi(\cdot | f(s))$ I wonder if this isn’t a method to allow prior knowledge to flow into the policy and value estimation. If the transformation(s) $f_i$ have been chosen such that one can be reasonably sure that true value and policy functions should be invariant to said transformations, then by enforcing $V(s) = V(f(s))$ and $\pi(\cdot | s) = \pi(\cdot | f(s))$  one is constraining V and \pi to fit the prior knowledge contained in $f_i$.

So now we’re comparing apples and oranges, since your method (DrAC) gets to incorporate this prior knowledge, while PPO and RAD don’t, and DrAC’s good performance isn’t surprising.

Automatic Data Augmentation
----------------------------------------

Here, given some candidates for data augmentation, the authors propose three methods to discover which candidate work well. Unfortunately, I don’t fully understand the approach.

The authors are examining a Meta-Reinforcement Learning setting, where one wishes to find a policy which performs well not just on one MDP, but on a whole distribution of MDPs. This leads to an inner-and-outer for-loop like setting, in the inner for loop, the agent does multiple episodes with a single environment, in the outer loop the agent gets new environments.

I had expected this inner-outer for-loop structure to pe present in the meta learning of augmentations, and for the authors to clearly describe how knowledge about the effectiveness of augmentations is transferred from past episodes on one environment to future episodes in the same environment, and how it’s transferred from past environments to future environments.

The only support given for these methods is the experimental results. Yet we see that the performance of DrAC and UCB-DrAC lie within a standard deviation of one another. So the evidence of the effectiveness of UCB-DrAC over the simpler DrAC is weak at best.

Minor Comments:
-------------------------
$T_m(s’|s,a)$ is the transition function, $R_m(s,a)$ is the reward function: Here, $T$ and $R$ are generally distributions and not functions

Eq. 6 is confusing, since $f_*$ can refer to both $f_t$ (function at timestep $t$) and $f_i$ (the $i$-th transformation function).

Do the update with something like $N_t(f) \leftarrow N_{t-1}(f) + 1$ is a bit clearer, since then it’s the number of times $f$ has been pulled before timestep $t$

---

> ### Author Response · Authors · 2020-11-12
> **Thank you for your feedback!**
>
> We thank the reviewer for their feedback and we especially appreciate that the reviewer finds our writing clear and believes that data augmentation for reinforcement learning is an important research direction. We respond to the reviewer’s concerns below.
>
> Policy and Value Function Regularization
>
> C1: “One just needs to save the old policies (and old Advantage function), so that πθold can be applied to transformed states, and not just use the actions from the buffer. This would seem like a straightforward fix.”
>
> A1: We respectfully disagree with this comment as we believe such a solution would be incorrect and would not solve the problem we emphasize. Please note that the correct estimate of the policy gradient objective used in PPO is the one in equation (1) which does not use the augmented observations at all since we are estimating advantages for the actual observations A(s, a). The probability distribution used to sample advantages is \pi_{old}(a | s) (rather than \pi_{old}(a | f(s)) since we can only interact with the environment via the true observations and not the augmented ones (because the reward and transition functions are not defined in this case). Hence, the correct importance sampling estimate uses \pi(a|s) / \pi_{old}(a | s). If we understood correctly, what you propose is to use  \pi(a | f(s)) / \pi_{old}(a | f(s)), but this is incorrect for the reasons mentioned above. What we argue is that, in the case of RAD, the only way to use the augmented observations f(s) is in the policy gradient objective, whether by  \pi(a | f(s)) / \pi_{old}(a | f(s)) as you propose or  \pi(a | f(s)) / \pi_{old}(a | s) as RAD uses, but both are incorrect. In contrast, DrAC does not change the policy gradient objective at all which remains the one in equation (2) and instead uses the augmented observations in the additional regularization losses, as shown in equations (3), (4), and (5).
>
> C2: “So now we’re comparing apples and oranges, since your method (DrAC) gets to incorporate this prior knowledge, while PPO and RAD don’t, and DrAC’s good performance isn’t surprising.”
>
> A2: We respectfully disagree with this interpretation of our work. DrAC does not use any extra knowledge compared to RAD since they both use augmented observations to regularize RL agents. The only difference is that DrAC makes better use of this additional data by explicitly regularizing the policy and value function, while RAD attempts to do this implicitly. Please note that, even if RAD uses additional data, its performance can still be worse than PPO’s, while our method is comparable or better than PPO when using the same extra data as RAD (as shown in Figure 2).
>
> Automatic Data Augmentation
>
> C3: “I had expected this inner-outer for-loop structure to pe present in the meta learning of augmentations, and for the authors to clearly describe how knowledge about the effectiveness of augmentations is transferred from past episodes on one environment to future episodes in the same environment, and how it’s transferred from past environments to future environments.”
>
> A3: We will add more details about the meta-learning training. Please note that we have also included the code in our submission and we plan to open source it for full transparency.
>
> C4: “Yet we see that the performance of DrAC and UCB-DrAC lie within a standard deviation of one another. So the evidence of the effectiveness of UCB-DrAC over the simpler DrAC is weak at best.”
>
> A4: We never claim that the performance of UCB-DrAC is better than that of DrAC. In fact, DrAC with the best augmentation is an upper bound to UCB-DrAC since DrAC uses the best augmentation during the entire training process while UCB-DrAC needs to first find the best augmentation from a given set. The advantage of UCB-DrAC over DrAC is not superior performance but rather the fact that it provides an automatic way of selecting a good augmentation for a given task. As explained in section 3.3, UCB-DrAC reduces the computational resources needed to train an RL agent with data augmentation since it only needs to train a model once, while DrAC needs to train models with each type of augmentation in order to select the best one. Our point is that UCB-DrAC still achieves comparable performance with DrAC trained with the best augmentation, as shown in Figure 3.
>
> C5: Minor Comments.
>
> A5: Thank you for pointing these out. We will change our notation accordingly.
>
> In light of these clarifications, we would appreciate it if the reviewer confirmed that all their concerns have been addressed and, if so, reconsider their assessment.

---

> > ### Comment · AnonReviewer4 · 2020-11-24
> > **I thank the authors for the paper updates and clarifying comments.**
> >
> > After considering the authors response, I find my concern that the authors haven’t demonstrated why their method works still stands. They claim they’re fixing an issue with Laskin’s work, but don’t provide evidence that the issue in Laskin’s work is the real problem, nor do they address other fairly plausible explanations why the method could work on the experiments they chose. So I still argue for reject, but now with the modifications and clarifying comments I'm less forcefully in favor of rejection
> >
> > C1: “One just needs to save the old policies (and old Advantage function), so that $\pi_{\theta_\text{old}}$ can be applied to transformed states, and not just use the actions from the buffer. This would seem like a straightforward fix.”
> >
> > Thank you for you answer, I think I understand the issue more clearly now. You claim the poor performance is as a result of the inaccurate estimates resulting from $\pi(\cdot\mid s) \not= pi(\cdot\mid f(s))$.
> >
> > Yet it would seem that the $\pi(\cdot\mid s)\not=pi(\cdot\mid f(s))$ issue is fixed by regularizing the $G_\pi$ term, yet in your work you regularize both $G_\pi$ and $G_V$, and there’s nothing in the paper showing why $G_V$ is important or showing what happens when one regularizes only one of the two terms. Which brings me back to the original concern, which is that it seems like the authors discard straightforward fixes to the issue in favor of their method, which seem to work well in experiments but I suspect it works for another reason.
> >
> > C2: “So now we’re comparing apples and oranges, since your method (DrAC) gets to incorporate this prior knowledge, while PPO and RAD don’t, and DrAC’s good performance isn’t surprising.”
> >
> > Yet again, I thank the authors for their answer. Indeed the authors are correct when they write “DrAC does not use any extra knowledge compared to RAD since they both use augmented observations to regularize RL agents.” On this point I stand corrected.
> >
> > Still, the paper provides no insight into how much performance is gained by using non-augmented observations while regularizing the policy and value functions such that $V(s)=V(f(s))$ and similar for $\pi$. Therefore, since Laskin uses data augmentation and here the authors use data augmentation  (without showing how important that component is) combined with regularization of the value and policy, I still think we could be comparing apples and oranges; comparing a data augmentation method and a regularization method which forces $\pi$ and $V$ to conform to pre-existing knowledge of invariance, and that the method works for reasons other than that which the authors describe.
> >
> > C3: I thank the authors for adding Alg. 2 to the appendix. This greatly helps the reader  understand your approach. Something that would help further would be to clarify what the “update the policy and value function” on line 7 means. These functions were not defined before in the algorithm, so it’s unclear if they are defined in Alg. 1, used and then thrown away or if they somehow persist on in Alg. 2.
> >
> > C4: “Yet we see that the performance of DrAC and UCB-DrAC lie within a standard deviation of one another. So the evidence of the effectiveness of UCB-DrAC over the simpler DrAC is weak at best.”
> >
> > A4: We never claim that the performance of UCB-DrAC is better than that of DrAC.
> >
> > I see, fair enough.
> >
> > Minor Comment:
> >
> > $J_{\text{AC}}$ in line 13 of Alg. 1 is not defined. Plus, in Eq. 5 and line 13 of Alg. 1, $J_{\text{DrAC}}$ gets two different definitions. I would recommend keeping the definition of $J_{\text{DrAC}}$ consistent and defining $J_{\text{AC}}$ somewhere.

---

> > > ### Author Response · Authors · 2020-11-24
> > > **Response to Additional Comments (part 1)**
> > >
> > > Thank you for responding to our comments and engaging in the discussion. We believe there are some misunderstandings which we seek to clarify here in the hope that you will feel fully comfortable supporting our paper.
> > >
> > > C1: “Yet it would seem that the $\pi(⋅∣s)≠\pi(⋅∣f(s))$ issue is fixed by regularizing the $G_\pi$ term, yet in your work you regularize both $G_\pi$ and $G_V$, and there’s nothing in the paper showing why
> > > $G_V$ is important or showing what happens when one regularizes only one of the two terms.”
> > >
> > > A1: The regularization of $G_V$ is important in itself for learning a value function that is invariant to certain transformations of the observation. In addition, the learned value function is used to compute the generalized advantage estimate, which is then used to update the policy. Hence, regularizing the value function also indirectly influences the learned policy. As we argue in the paper, learning invariant policies and value functions can be useful for generalizing to new scenarios. Imagine that at test time, you encounter similar observations from training but with different backgrounds (e.g. different colors). In this case, you would like your value function to predict similar expected returns for two observations with the same underlying state but different backgrounds, which can be achieved via training with DrAC with random-conv or color-jitter augmentations. While we would be happy to run an ablation showing the effects of $G_\pi$ and $G_V$ separately, please note that this is simply unfeasible in the remaining discussion time which ends in a few hours. We are counting on your good faith to trust that we will run these experiments to even more thoroughly evaluate our method and that you will take this into account when making a final recommendation for our paper. Given that this request was only mentioned in the last few hours of the discussion period, there is no way for us to better address it at this point.
> > >
> > >
> > > C2: “Which brings me back to the original concern, which is that it seems like the authors discard straightforward fixes to the issue in favor of their method, which seem to work well in experiments but I suspect it works for another reason.”
> > >
> > > A2: Could you please explain what other reasons you have in mind? We are unsure what other fixes you would consider to be “straightforward” and why such fixes would be preferable to our proposed approach. While further analysis of why a certain method works is always welcome, we believe this is outside the scope of our paper. We have proposed a novel, simple, and effective method and shown undeniable gains over the recently published approach by Laskin et al. 2020, as well as over other strong baselines and ablations, on two challenging benchmarks for generalization in reinforcement learning.
> > >
> > >
> > > C3: “Still, the paper provides no insight into how much performance is gained by using non-augmented observations while regularizing the policy and value functions such that $V(s)=V(f(s))$ and similar for $\pi$.”
> > >
> > > A3: I think there might be some misunderstanding because this is exactly what we do -- we show how much performance is gained by using non-augmented observations in the buffer while regularizing the policy and value functions. The PPO buffer contains only non-augmented observations, but we use augmented observations to regularize the policy and value function such that $V(s) = V(f(s))$ and $\pi(s) = \pi(f(s))$.
> > >
> > >
> > > C4: “Therefore, since Laskin uses data augmentation and here the authors use data augmentation (without showing how important that component is) combined with regularization of the value and policy, I still think we could be comparing apples and oranges; comparing a data augmentation method and a regularization method which forces \pi and V to conform to pre-existing knowledge of invariance, and that the method works for reasons other than that which the authors describe.”
> > >
> > > A4: We believe we have empirically shown how important the regularization component is by comparing DrAC with RAD and demonstrating that DrAC significantly outperforms RAD, by a wide margin in certain cases (see Figure 2). RAD uses data augmentation as part of the PPO buffer while DrAC uses data augmentation as part of the regularization terms. We chose to compare DrAC with RAD since, as far as we know, RAD is the closest method to DrAC. We are unsure what you would consider to be a more convincing experiment to support our claim. We are also unsure what you mean by “the method works for reasons other than that which the authors describe”. What other reasons do you have in mind and how do you recommend we disentangle the different factors? We are happy to address any concerns you might have if you can provide guidance on how we can alleviate them.

---

> > > > ### Comment · AnonReviewer4 · 2020-11-24
> > > > **Some clarifying remarks on my last response**
> > > >
> > > > C3: "I think there might be some misunderstanding because this is exactly what we do"
> > > >
> > > > A3: I see, yes there was a misunderstanding, I appologize. I though that the augmentation would happen in $J_\text{PPO}$ on line 13 of Alg. 1. Although the authors didn't mention whether augmentation happens here or not, I'd assumed this was the case since data augmentation is in the name of the paper and features prominently throughout. If then no data augmentation happens in $J_\text{PPO}$, the only place the transformation function $f$ is applied is in the regularization terms $G_\pi$ and $G_V$. And the paper is about a method for regularizing $\pi$ and $V$ to conform to pre-existing knowledge of invariance, and isn't so much about data augmentation.
> > > >
> > > > Which brings me to:
> > > >
> > > > C2.1: "Could you please explain what other reasons you have in mind?"
> > > >
> > > > A2.1: Yes, a fair enough request. It would seem to me that the method the authors propose works, but not for the reasons given in the theory section. There, they argue they fix an issue with Laskin's data augmentation while it seems to me that their approach is a pure regularize-to-incorporate-prior knowledge approach. It could be (I'm not sure, but I strongly suspect) this method works better than Laskin's since there the only thing pushing $V$ and $\pi$ to be invariant is the augmented data. This is a somewhat "weak" signal, as it must work its way through many updates of the value function to reach $\pi$. The author's approach, on the other hand, directly regularizes $\pi$ and $V$, thus giving a much stronger transformation invariance signal to $\pi$.
> > > >
> > > > Which nicely dovetails with your statement:
> > > >
> > > > C1: The regularization of $G_V$ is important in itself for learning a value function
> > > >
> > > > A1: Yes, I think we can agree on this. I also apologize that I didn't respond sooner (I fell ill, no COVID thankfully). This situation and my untimely response is unfair to the authors. Still, I'm not sure how much an extra week or two would have helped as much work lies in front of the authors to either back up their theory with evidence or rework their theory to be more easily justifiable.
> > > >
> > > > C2.2: "We are unsure what other fixes you would consider to be “straightforward” and why such fixes would be preferable to our proposed approach"
> > > >
> > > > For example, just using the $G_\pi$ regularization term without $G_V$ is a straightforward fix to the $\pi(\cdot\mid s)\not=\pi(\cdot\mid f(s))$ issue. If indeed, this issue is the reason why Laskin's method fails, then this fix should suffice and would be preferable since it's simpler.
> > > >
> > > > I strongly suspect (and I believe the authors concur) that also using the $G_V$ term improves performance. But the question is why, when this would seem to contradict the theoretical justification which is that $\pi(\cdot\mid s)\not=\pi(\cdot\mid f(s))$ is the issue with Laskin's method.
> > > >
> > > > C2.3: "While further analysis of why a certain method works is always welcome, we believe this is outside the scope of our paper. We have proposed a novel, simple, and effective method and shown undeniable gains over the recently published approach by Laskin et al. 2020..."
> > > >
> > > > A2.3: On the novel, simple and effective point I can agree. My issue is that the authors also propose a theory for why this approach works, but do not provide evidence supporting this theory. If a paper proposes a theory why something works, then providing evidence why the theory is correct should be very much within the scope of the paper. Further, I have my doubts on the soundness of this theory (which I discussed above).
> > > >
> > > > C4: "We chose to compare DrAC with RAD since, as far as we know, RAD is the closest method to DrAC"
> > > >
> > > > A4: I think the comparison to RAD is justified and convincing in the sense that I believe DrAC does indeed work better than RAD. Its just I doubt the theoretical justification, and the theory isn't strongly supported by evidence.

---

> > > > > ### Author Response · Authors · 2020-11-24
> > > > > **Follow-up Response**
> > > > >
> > > > > Thank you for your quick response and for the clarifications. We are glad to see that you agree with us on most points and we believe we now better understand the crux of what is preventing you from supporting our paper. We still believe it is based on a misunderstanding regarding our contributions which we aim to explain below.
> > > > >
> > > > > It is our understanding that the main issue preventing you from recommending acceptance of our paper is the fact that it is not crystal clear from our experiments whether the gains of DrAC over RAD are due to fixing the theoretical problem with RAD we emphasize in Section 3.1 or due to providing a stronger signal via the regularization of the policy and value function. However, it seems that you agree this could be tested by running an ablation that only uses $G_\pi$ to disentangle the effect of the policy regularization (which would fix the problem we point out regarding RAD) from the effect of regularizing both the policy and the value function (which would provide an even stronger invariance signal). This can be done and could have likely been done during the discussion period if we had been aware about this concern.
> > > > >
> > > > > However, we believe that our claims and results still hold even if we do not know exactly where the gains are coming from. The central claim of our paper is that regularizing the policy and value function improves test performance on RL tasks. Hence, we do not see why also using value regularization alongside policy regularization would be a problem if this helps and makes sense in our view (for the reasons mentioned in our previous response regarding that invariance in both the value and the policy is desirable for generalization in RL). A secondary claim is that a naive use of data augmentation in RL as done in RAD leads to certain theoretical problems. We believe we have already supported this claim with our formal explanation in Section 3 and Appendix B and that this could be further supported with empirical experiments using only $G_\pi$, which we plan to perform.
> > > > >
> > > > > C1: "Still, I'm not sure how much an extra week or two would have helped as much work lies in front of the authors to either back up their theory with evidence or rework their theory to be more easily justifiable."
> > > > > A1: We are sorry to hear about your falling ill. However, we respectfully disagree and we do believe we would have been able to run the additional experiments required to convince you that our claim is well supported by empirical evidence. Alternatively, we could have simply changed some of the wording of the paper (i.e. saying that we make two separate contributions 1) emphasize a theoretical problem with RAD and 2) propose a method for regularizing the policy and value function that does not have this problem without explicitly saying that our method directly fixes RAD's problem) as you also suggested which would have been a minor fix given a reasonable amount of time. We would be happy to rephrase our claims if this makes you more comfortable in supporting our paper.
> > > > >
> > > > > C2: "I strongly suspect (and I believe the authors concur) that also using the  term improves performance. But the question is why, when this would seem to contradict the theoretical justification which is that  is the issue with Laskin's method."
> > > > > A2: We respectfully disagree on this point. We do not believe this would contradict the theoretical justification at all. If only using $G_\pi$ is better than RAD, but using both $G_\pi$ and $G_V$ is better than only using $G_\pi$, then our claim about the problem with RAD still holds and we show additional benefits from also regularizing the value function.
> > > > >
> > > > > Finally, we do not find a score of 4 to be fully consistent with (at least our understanding of) your current review. While we fully understand the importance of running this ablation study and we believe including them would strengthen our paper (for which we thank you for pointing it out), we believe this is a rather minor issue given that our main claims and results stand even without this ablation.

---

> > > ### Author Response · Authors · 2020-11-24
> > > **Response to Additional Comments (part 2)**
> > >
> > > C5: “Something that would help further would be to clarify what the “update the policy and value function” on line 7 means.“
> > >
> > > A5: We have updated our paper with definitions for the policy and value function in Algorithms 2, 3 and 4, which we hope clarifies your question. The policy and value functions are updated at each interaction in Algorithms 2, 3, and 4, using one DrAC update as described in Algorithm 1. For simplicity, we initially left these details out since it is explained in the text that the agent’s policy and value functions are trained (using DrAC from Algorithm 1) alongside the methods for automatically selecting an augmentation (i.e. UCB, RL2, and Meta which are described in detail in Algorithms 2, 3, and 4).
> > >
> > > Please note that we have also submitted the code along with instructions for how to run it and we plan to open source it. We hope this will allow others to reproduce the results and understand exactly how the algorithms are implemented.
> > >
> > > C6: Minor Comment
> > > A6: We fixed our notation to be consistent and fully defined in the paper.
> > >
> > > Cognizant that there are only a few hours left to complete the discussion, we have tried to promptly respond to your concerns and have updated the paper to fix some minor notation issues, as you requested. However, we hope that in your final recommendation you will take into account the fact that it was simply not possible for us to run further experiments in the last couple hours of the discussion period.
> > >
> > > We believe our paper introduces a new method, thoroughly compares it against other strong baselines on two challenging and widely used benchmarks, and establishes a new state of the art. While we agree there is always room for more extensive analysis, we ask you to consider that we have put significant effort towards comparing against related methods and relevant ablations. We are confident from our discussion we have made the case to convince you that "4" is too low a grade, and we hope that you are willing to change your mind and decide to strongly support our publication. Please note that the scale of 1-10 leaves room for such assessments, e.g. 7 or 8, which both support the paper but state there is some room for improvement. We hope you will agree, and we thank you again for your valuable feedback and for engaging in the discussion process.

---

### Official Review · AnonReviewer1 · 2020-10-26
**Interesting results**

**Rating:** 7
**Confidence:** 3

**Review:**

This paper presents a method that utilizes data augmentation for image-based reinforcement learning. The data augmentation is used to regularize the policy and function approximation in the proposed method. In addition, a method for automatically identifying effective ways of data augmentation is proposed. The experimental results show that the proposed method outperforms the baseline methods.

The study shows that the regularization of policy and function approximation using the transformed images is more effective than training a policy by using the transformed image as a state. Regarding identification of effective ways of data augmentation,  the proposed method seems improve the performance of image-based RL methods, although the proposed approach is simple.

I have some suggestions for improving the paper.
- The term for regularizing the policy is given in Eq. (3). I think that the first \pi(a|s) in the KL divergence should also have the subscript \theta because it seems that the two policies are based on the same model. Likewise, the first term in Eq. (4) should have the same subscript as the second term.

- It would be good to show the Jensen-Shannon divergence and the cycle consistency of RAD in Table 2.


== comments after discussions and the paper update ==

I appreciate the authors' efforts to improve the clarity and provide additional results. I believe that the proposed method is now clearly presented and the claims are properly supported by experiments. I raise the score to "accept".

---

> ### Author Response · Authors · 2020-11-12
> **Thank you for your feedback!**
>
> We thank the reviewer for their encouraging and constructive feedback. Below we would like to respond to the reviewer’s suggestions for improving our work.
>
> C1: “The term for regularizing the policy is given in Eq. (3). I think that the first \pi(a|s) in the KL divergence should also have the subscript \theta because it seems that the two policies are based on the same model. Likewise, the first term in Eq. (4) should have the same subscript as the second term.”
>
> A1: It is true that the two policies in the equation, \pi(a|s) and  \pi(a|f(s, \nu)) have the same parameters \theta. The reason we left out the \theta subscript from \pi(a|s) is to indicate that we are only backpropagating gradients through  \pi(a|f(s, \nu)). This is mentioned right below equation (5). But we appreciate the suggestion and we will make our notation more clear.
>
> C2: “It would be good to show the Jensen-Shannon divergence and the cycle consistency of RAD in Table 2.”
>
> A2: We agree this would be valuable and we will add these results in the updated draft.
>
> In light of the promised changes, we would appreciate it if the reviewer would be willing to reconsider their assessment and/or provide us with further feedback.

---

### Public Comment · ~Bogdan_Mazoure1 · 2020-11-15
**Related works**

I think this is an interesting paper on one additional form of data augmentation for generalization in RL, which might greatly simplify existing data aug. approaches by making the process automatic. I believe the recent papers $^{1,2}$ would be a great addition to the related works section since they also touch upon data augmentation and generalization in RL.
$^1$. Stooke, Adam, et al. "Decoupling representation learning from reinforcement learning." arXiv preprint arXiv:2009.08319 (2020).,
$^2$. Mazoure, Bogdan, et al. "Deep reinforcement and infomax learning." Advances in Neural Information Processing Systems 33 (2020).

---

> ### Author Response · Authors · 2020-11-15
> **Thank you for the suggestion**
>
> Thank you for these suggestions. We agree they are relevant and we will add them to our related work.

---

### Author Response · Authors · 2020-11-21
**Paper Update**

We would like to thank you again for your constructive feedback which has helped us further improve our paper. We have uploaded a revised draft which contains the following updates:

*. Results on the DeepMind Control Suite with distractor backgrounds (Section 4.5 and Appendix K), where our method still outperforms the baselines.

*. Pseudocodes and more details for all our proposed methods (Section 3.2 and Appendix D).

*. Comparison with RAD in the robustness analysis (Section 4.4).

*. Explained in greater detail why a naive application of data augmentation in RL can be problematic (Appendix B).

*. Clarified the notation in Sections 3.2 and 3.3.

We hope these changes, together with our responses below, address your concerns and are sufficient for you to reconsider your assessment of the paper. If you have any remaining concerns, please do not hesitate to let us know so we can discuss them.

---

### Decision · Program_Chairs · 2021-01-07
**Final Decision**

**Decision:**

Reject

**Comment:**

As of now, automatic data augmentation methods have mostly been proposed for supervised learning tasks, especially classification. This paper introduces automatic data augmentation to deep (image-based) reinforcement learning agents, aiming to make the agents generalize better to new environments. A new algorithm called data-regularized actor-critic (DrAC) is proposed, with three variants that correspond to different methods for automatically finding a useful augmentation: UCB-DrAC, RL2-DrAC, and Meta-DrAC. Promising results are reported on OpenAI’s Procgen generalization benchmark which consists of 16 procedurally generated environments (games) with visual observations. Further experiments have been added in the revised version.

**Strengths:**
  * This work is among the first attempts that propose an automatic data augmentation scheme for reinforcement learning.
  * The paper articulates well the problem of data augmentation for reinforcement learning.
  * The experiment results are generally promising.

**Weaknesses:**
  * Although the experiment results reported seem promising, there are missing pieces in order to help the readers gain a deeper understanding to justify more thoroughly why the proposed regularization-based scheme works.
  * Theoretical justification is lacking.

This is a borderline paper. While it presents some interesting ideas supported empirically by experiment results, the paper in its current form is premature for acceptance since a more thorough, scientific treatment is lacking before drawing conclusions. Moreover, considering that there are many competitive submissions to ICLR, I do not recommend that the paper be accepted. Nevertheless, the authors are encouraged to address the concerns raised to fill the gaps when revising their paper for resubmission in the future.